# Pure Exploration with Multiple Correct Answers

**Rémy Degenne**
Centrum Wiskunde & Informatica
Science Park 123, Amsterdam, NL
`remy.degenne@cwi.nl`

**Wouter M. Koolen**
Centrum Wiskunde & Informatica
Science Park 123, Amsterdam, NL
`wmkoolen@cwi.nl`

## Abstract

We determine the sample complexity of pure exploration bandit problems with multiple good answers. We derive a lower bound using a new game equilibrium argument. We show how continuity and convexity properties of *single-answer* problems ensure that the existing Track-and-Stop algorithm has asymptotically optimal sample complexity. However, that convexity is lost when going to the *multiple-answer* setting. We present a new algorithm which extends Track-and-Stop to the multiple-answer case and has asymptotic sample complexity matching the lower bound.

## 1   Introduction

In *pure exploration* aka *active testing* problems the learning system interacts with its environment by sequentially performing experiments to quickly and reliably identify the answer to a particular pre-specified question. Practical applications range from simple queries for cost-constrained physical regimes, i.e. clinical drug testing, to complex queries in structured environments bottlenecked by computation, i.e. simulation-based planning. The theory of pure exploration is studied in the multi-armed bandit framework. The scientific challenge is to develop tools for characterising the sample complexity of new pure exploration problems, and methodologies for developing (matching) algorithms. With the aim of understanding power and limits of existing methodology, we study an extended problem formulation where each instance may have multiple correct answers. We find that multiple-answer problems present a phase transition in complexity, and require a change in our thinking about algorithms.

The existing methodology for developing asymptotically instance-optimal algorithms, Track-and-Stop by Garivier and Kaufmann [2016], exploits the so-called *oracle weights*. These probability distributions on arms naturally arise in sample complexity lower bounds, and dictate the optimal sampling proportions for an "oracle" algorithm that needs to be sample efficient only on exactly the current problem instance. The main idea is to track the oracle weights computed at a converging estimate of the instance. The analysis of Track-and-Stop requires continuity of the oracle weights as a function of the bandit model. For the core Best Arm Identification problem, discontinuity only occurs at degenerate instances where the sample complexity explodes. So this assumption seemed harmless.

**Our contribution**   We show that the oracle weights may be non-unique, even for single-answer problems, and hence need to be regarded as a set-valued mapping. We show this mapping is always (upper hemi-)continuous. We show that each instance maps to a convex set for single-answer problems, and this allows us to extend the Track-and-Stop methodology to all such problems. At instances with non-singleton set-valued oracle weights more care is needed: of the two classical tracking schemes "C" converges to the convex set, while "D" may fail entirely.

We show that for multiple-answer problems convexity is violated. There are instances where two distinct oracle weights are optimal, while no mixture is. Unmodified tracking converges in law

(experimentally) to a distribution on the full convex hull, and suffers as a result. We propose a "sticky" modification to stabilise the approach, and show that now it converges to only the corners. We prove that Sticky Track-and-Stop is asymptotically optimal.

**Related work**   Multi-armed bandits have been the subject of intense study in their role as a model for medical testing and reinforcement learning. For the objective of reward maximisation [Berry and Fristedt, 1985, Lai and Robbins, 1985, Auer et al., 2002, Bubeck and Cesa-Bianchi, 2012] the main challenge is balancing exploration and exploitation. The field of pure exploration (active testing) focuses on generalisation vs sample complexity, in the fixed confidence, fixed budget and simple regret scalarisations. It took off in machine learning with the *multiple-answer* problem of $(\epsilon, \delta)$-Best Arm Identification (BAI) [Even-Dar et al., 2002]. Early results focused on worst-case sample complexity guarantees in sub-Gaussian bandits. Successful approaches include *Upper and Lower confidence bounds* [Bubeck et al., 2011, Kalyanakrishnan et al., 2012, Gabillon et al., 2012, Kaufmann and Kalyanakrishnan, 2013, Jamieson et al., 2014], *Racing or Successive Rejects/Eliminations* [Maron and Moore, 1997, Even-Dar et al., 2006, Audibert et al., 2010, Kaufmann and Kalyanakrishnan, 2013, Karnin et al., 2013].

Fundamental information-theoretic barriers [Castro, 2014, Kaufmann et al., 2016, Garivier and Kaufmann, 2016] for each specific problem instance refined the worst-case picture, and sparked the development of instance-optimal algorithms for single-answer problems based on *Track-and-Stop* [Garivier and Kaufmann, 2016] and *Thompson Sampling* [Russo, 2016]. For multiple-answer problems the elegant KL-contraction-based lower bound is not sharp, and new techniques were developed by Garivier and Kaufmann [2019].

Recent years also saw a surge of interest in pure exploration with *complex queries* and *structured environments*. Kalyanakrishnan and Stone [2010] identify the top-$M$ set, Locatelli et al. [2016] the arm closest to a threshold, and Chen et al. [2014], Gabillon et al. [2016] the optimiser over an arbitrary combinatorial class. For arms arranged in a matrix Katariya et al. [2017] study BAI under a rank-one assumption, while Zhou et al. [2017] seek to identify a Nash equilibrium. For arms arranged in a minimax tree there is significant interest in finding the optimal move at the root Teraoka et al. [2014], Garivier et al. [2016], Huang et al. [2017], Kaufmann and Koolen [2017], Kaufmann et al. [2018], as a theoretical model for studying Monte Carlo Tree search (which is a planning sub-module of many advanced reinforcement learning systems).

## 2   Notations

We work in a given one-parameter one-dimensional canonical exponential family, with mean parameter in an open interval $\mathcal{O} \subseteq \mathbb{R}$. We denote by $d(\mu, \lambda)$ the KL divergence from the distribution with mean $\mu$ to that with mean $\lambda$. A $K$-armed bandit model is identified by its vector $\boldsymbol{\mu} \in \mathcal{O}^K$ of mean parameters. We denote by $\mathcal{M} \subseteq \mathcal{O}^K$ the set of possible mean parameters in a specific bandit problem. Excluding parts of $\mathcal{O}^K$ from $\mathcal{M}$ may be used to encode a known structure of the problem. We assume that there is a finite domain $\mathcal{I}$ of answers, and that the *correct answer* for each bandit model is specified by a set-valued function $i^* : \mathcal{M} \to 2^{\mathcal{I}}$.

**Example 1.** As our running example, we will use the *Any Low Arm* multiple-answer problem. Given a threshold $\gamma \in \mathcal{O}$, the goal is return the index $k$ of any arm with $\mu_k < \gamma$ if such an arm exists, or to return "no" otherwise. Formally, we have possible answers $\mathcal{I} = [K] \cup \{\text{no}\}$, and correct answers

$$i^*(\boldsymbol{\mu}) \;=\; \begin{cases} \{k \mid \mu_k \le \gamma\} & \text{if } \min_k \mu_k < \gamma, \\ \{\text{no}\} & \text{if } \min_k \mu_k > \gamma. \end{cases}$$

We exclude the case $\min_k \mu_k = \gamma$ from $\mathcal{M}$ (as such instances require infinite sample complexity).

Additional examples of multiple-answer identification problems are visualised in Table 1 in Appendix B.

**Algorithms.**   A learning strategy is parametrised by a stopping rule $\tau_\delta \in \mathbb{N}$ depending on a parameter $\delta \in [0, 1]$, a sampling rule $A_1, A_2, \ldots \in [K]$, and a recommendation rule $\hat{\imath} \in \mathcal{I}$. When a learning strategy meets a bandit model $\boldsymbol{\mu}$, they interactively generate a history $A_1, X_1, \ldots, A_\tau, X_\tau, \hat{\imath}$, where $X_t \sim \mu_{A_t}$. We allow the possibility of non-termination $\tau_\delta = \infty$, in which case there is no recommendation $\hat{\imath}$. At stage $t \in \mathbb{N}$, we denote by $N_t = (N_{t,1}, \ldots, N_{t,K})$ the number of samples (or "pulls") of the arms, and by $\hat{\boldsymbol{\mu}}_t$ the vector of empirical means of the samples of each arm.

**Confidence and sample complexity.**   For confidence parameter $\delta \in (0, 1)$, we say that a strategy is $\delta$-correct (or $\delta$-PAC) for bandit model $\boldsymbol{\mu}$ if it recommends a correct answer with high probability, i.e. $\mathbb{P}_{\boldsymbol{\mu}}\big(\tau_\delta < \infty \text{ and } \hat{\imath} \in i^*(\boldsymbol{\mu})\big) \geq 1 - \delta$. We call a given strategy $\delta$-correct if it is $\delta$-correct for every $\boldsymbol{\mu} \in \mathcal{M}$. We measure the statistical efficiency of a strategy on a bandit model $\boldsymbol{\mu}$ by its *sample complexity* $\mathbb{E}_{\boldsymbol{\mu}}[\tau_\delta]$. We are interested in $\delta$-correct algorithms minimizing sample complexity.

**Divergences.**   For any answer $i \in \mathcal{I}$, we define the *alternative to $i$*, denoted $\neg i$, to be the set of bandit models on which answer $i$ is incorrect, i.e.

$$\neg i := \{\boldsymbol{\mu} \in \mathcal{M} | i \notin i^*(\boldsymbol{\mu})\} \ .$$

We define the ($\boldsymbol{w}$-weighted) divergence from $\boldsymbol{\mu} \in \mathcal{M}$ to $\boldsymbol{\lambda} \in \mathcal{M}$ or $\Lambda \subseteq \mathcal{M}$ by

$$D(\boldsymbol{w}, \boldsymbol{\mu}, \boldsymbol{\lambda}) \;=\; \sum_k w_k d(\mu_k, \lambda_k) \qquad\qquad D(\boldsymbol{w}, \boldsymbol{\mu}, \Lambda) \;=\; \inf_{\boldsymbol{\lambda} \in \Lambda} D(\boldsymbol{w}, \boldsymbol{\mu}, \boldsymbol{\lambda})$$

$$D(\boldsymbol{\mu}, \Lambda) \;=\; \sup_{\boldsymbol{w} \in \triangle_K} D(\boldsymbol{w}, \boldsymbol{\mu}, \Lambda) \qquad\qquad D(\boldsymbol{\mu}) \;=\; \max_{i \in \mathcal{I}} D(\boldsymbol{\mu}, \neg i)$$

Note that $D(\boldsymbol{w}, \boldsymbol{\mu}, \Lambda) = 0$ whenever $\boldsymbol{\mu} \in \Lambda$. We denote by $i_F(\boldsymbol{\mu})$ the argmax (set of maximisers) of $i \mapsto D(\boldsymbol{\mu}, \neg i)$, and we call $i_F(\boldsymbol{\mu})$ the *oracle answer(s)* at $\boldsymbol{\mu}$. We write $\boldsymbol{w}^*(\boldsymbol{\mu}, \neg i)$ for the maximisers of $\boldsymbol{w} \mapsto D(\boldsymbol{w}, \boldsymbol{\mu}, \neg i)$, and call these the *oracle weights for answer $i$ at $\boldsymbol{\mu}$*. We write $\boldsymbol{w}^*(\boldsymbol{\mu}) = \bigcup_{i \in i_F(\boldsymbol{\mu})} \boldsymbol{w}^*(\boldsymbol{\mu}, \neg i)$ for the set of *oracle weights* among all oracle answers. We include expressions for the divergence when $i^*$ is generated by half-spaces, minima and spheres in Appendix H.

**Example 1** (Continued). Consider an *Any Low Arm* instance $\boldsymbol{\mu}$ with $\min_k \mu_k < \gamma$. For any arm $i \in i^*(\boldsymbol{\mu})$, we have $D(\boldsymbol{w}, \boldsymbol{\mu}, \neg i) = w_i d(\mu_i, \gamma)$ and $D(\boldsymbol{\mu}, \neg i) = d(\mu_i, \gamma)$. Hence $D(\boldsymbol{\mu}) = d(\min_i \mu_i, \gamma)$, and $i_F(\boldsymbol{\mu}) = \operatorname{argmin}_i \mu_i$. On the other hand, when $\min_k \mu_k > \gamma$, we have $i^*(\boldsymbol{\mu}) = \{\text{no}\}$, so that $D(\boldsymbol{w}, \boldsymbol{\mu}, \neg\text{no}) = \min_k w_k d(\mu_k, \gamma)$ and $D(\boldsymbol{\mu}, \neg\text{no}) = D(\boldsymbol{\mu}) = 1 \big/ \sum_{k=1}^K \frac{1}{d(\mu_k, \gamma)}$.

The function $i_F(\boldsymbol{\mu}) = \{i \in \mathcal{I} : D(\boldsymbol{\mu}, \neg i) = D(\boldsymbol{\mu})\}$ is set valued, as is $\boldsymbol{w}^*$. They are singletons with continuous value over some connected subsets of $\mathcal{M}$, and are multi-valued on common boudaries of two or more such sets. Both $i_F$ and $\boldsymbol{w}^*$ can be thought of as having single values, unless $\boldsymbol{\mu}$ sits on such a boundary, in which case we will prove that they are equal to the union (or convex hull of the union) of their values in the neighbouring regions.

## 3   Lower Bound

We show a lower bound for any algorithm for multiple-answer problems. Our lower bound extends the single-answer result of Garivier and Kaufmann [2016]. We are further inspired by Garivier and Kaufmann [2019], who analyse the $\epsilon$-BAI problem. They prove lower bounds for algorithms with a sampling rule independent of $\delta$, imposing the further restriction that either $K = 2$ or that the algorithm ensures that $N_{t,k}/t$ converges as $t \to \infty$. The new ingredient in this section is a game-theoretic equilibrium argument, which allows us to analyse any $\delta$-correct algorithm in any multiple answer problem. Our main lower bound is the following.

**Theorem 1.** *Any $\delta$-correct algorithm verifies*

$$\liminf_{\delta \to 0} \frac{\mathbb{E}_{\boldsymbol{\mu}}[\tau_\delta]}{\log(1/\delta)} \;\geq\; T^*(\boldsymbol{\mu}) := D(\boldsymbol{\mu})^{-1} \quad where \quad D(\boldsymbol{\mu}) \;=\; \max_{i \in i^*(\boldsymbol{\mu})} \max_{\boldsymbol{w} \in \triangle_K} \inf_{\boldsymbol{\lambda} \in \neg i} \sum_{k=1}^K w_k d(\mu_k, \lambda_k)$$

*for any multiple answer instance $\boldsymbol{\mu}$ with sub-Gaussian arm distributions.*

The proof is in Appendix C, where we also discuss how the convenient sub-Gaussian assumption can be relaxed. We would like to point out one salient feature here. To show sample complexity lower bounds at $\boldsymbol{\mu}$, one needs to find problems that are hard to distinguish from it statistically, yet require a different answer. We obtain these problems by means of a minimax result.

**Lemma 2.** *For any answer $i \in \mathcal{I}$, the divergence from $\boldsymbol{\mu}$ to $\neg i$ equals*

$$D(\boldsymbol{\mu}, \neg i) \;=\; \inf_{\mathbb{P}} \max_{k \in [K]} \mathbb{E}_{\boldsymbol{\lambda} \sim \mathbb{P}} \left[d(\mu_k, \lambda_k)\right].$$

*where the infimum ranges over probability distributions on $\neg i$ supported on (at most) $K$ points.*

The proof of Theorem 1 then challenges any algorithm for $\boldsymbol{\mu}$ by obtaining a witness $\mathbb{P}$ for $D(\boldsymbol{\mu}) = \max_i D(\boldsymbol{\mu}, \neg i)$ from Lemma 2, sampling a model $\boldsymbol{\lambda} \sim \mathbb{P}$, and showing that if the algorithm stops early, it must make a mistake w.h.p. on at least one model from the support. The equilibrium property of $\mathbb{P}$ allows us to control a certain likelihood ratio martingale regardless of the sampling strategy employed by the algorithm.

We discuss the novel aspect of Theorem 1 and its lessons for the design of optimal algorithms. First of all, for single-answer instances $|i^*(\boldsymbol{\mu})|{=}1$ we recover the known asymptotic lower bound [Garivier and Kaufmann, 2016, Remark 2]. For multiple-answer instances the bound says the following. The optimal sample complexity hinges on the *oracle answers* $i_F(\boldsymbol{\mu})$. That is, for $i_f \in i_F(\boldsymbol{\mu})$, the complexity of problem $\boldsymbol{\mu}$ is governed by the difficulty of discriminating $\boldsymbol{\mu}$ from the set of models on which answer $i_f$ is incorrect.

Is the bound tight? We argue yes. Consider the following oracle strategy, which is specifically designed to be very good at $\boldsymbol{\mu}$. First, it computes a pair $(i, \boldsymbol{w})$ witnessing the two outer maxima in $D(\boldsymbol{\mu})$. The algorithm samples according to $\boldsymbol{w}$. It stops when it can statistically discriminate $\hat{\boldsymbol{\mu}}_t$ from $\neg i$, and outputs $\hat{\imath} = i$. This algorithm will indeed have expected sample complexity equal to $D(\boldsymbol{\mu})^{-1}$ at $\boldsymbol{\mu}$, and it will be correct.

The above oracle viewpoint presents an idea for designing algorithms, following Garivier and Kaufmann [2016] and Chen et al. [2017]. Perform a lower-order amount of forced exploration of all arms to ensure $\hat{\boldsymbol{\mu}}_t \to \boldsymbol{\mu}$. Then at each time point compute the empirical mean vector $\hat{\boldsymbol{\mu}}_t$ and oracle weights $\boldsymbol{w}_t \in \boldsymbol{w}^*(\hat{\boldsymbol{\mu}}_t)$. Then sample according to $\boldsymbol{w}_t$. This approach is successful for single-answer bandits with unique and continuous oracle weights. We argue in Section 4.3 below that it extends to points of discontinuity by exploiting upper hemicontinuity and convexity of $\boldsymbol{w}^*$.

For multiple-answer bandits, we argue that the set of maximisers $\boldsymbol{w}^*(\boldsymbol{\mu})$ is no longer convex when $i_F(\boldsymbol{\mu})$ is not a singleton. It can then happen that $\hat{\boldsymbol{\mu}}_t \to \boldsymbol{\mu}$, while at the same time $\boldsymbol{w}^*(\hat{\boldsymbol{\mu}}_t)$ keeps oscillating. If the algorithm tracks $\boldsymbol{w}^*(\hat{\boldsymbol{\mu}}_t)$, its sampling proportions will end up in the convex hull of $\boldsymbol{w}^*(\boldsymbol{\mu})$. However, as $\boldsymbol{w}^*(\boldsymbol{\mu})$ is not convex itself, these proportions will not be optimal. We present empirical evidence for that effect in Appendix D. The lesson here is that the oracle needs to pick an answer and "stick with it". This will be the basis of our algorithm design in Section 5.

## 4 Properties of the Optimal Allocation Sets

The Track-and-Stop sampling strategy aims at ensuring that the sampling proportions converge to oracle weights. In the case of a singleton-valued oracle weights set $\boldsymbol{w}^*(\boldsymbol{\mu})$ for single answer problems, that convergence was proven in [Garivier and Kaufmann, 2016]. We study properties of that set with the double aim of extending Track-and-Stop to points $\boldsymbol{\mu}$ where $\boldsymbol{w}^*(\boldsymbol{\mu})$ is not a singleton and of highlighting what properties hold only for the single-answer case, but not in general.

### 4.1 Continuity

We first prove continuity properties of $D$ and $\boldsymbol{w}^*$. We show how the convergence of $\hat{\boldsymbol{\mu}}_t$ to $\boldsymbol{\mu}$ translates into properties of the divergences from $\hat{\boldsymbol{\mu}}_t$ to the alternative sets.

For a set $B$, let $\mathbb{S}(B) = 2^B \setminus \{\emptyset\}$ be the set of all *non-empty* subsets of $B$.

**Definition 3** (Upper hemicontinuity). A set-valued function $\Gamma : A \to \mathbb{S}(B)$ is upper hemicontinuous at $a \in A$ if for any open neighbourhood $V$ of $\Gamma(a)$ there exists a neighbourhood $U$ of $a$ such that for all $x \in U$, $\Gamma(x)$ is a subset of $V$.

**Theorem 4.** *For all $i \in \mathcal{I}$,*

1. *the function $(\boldsymbol{w}, \boldsymbol{\mu}) \mapsto D(\boldsymbol{w}, \boldsymbol{\mu}, \neg i)$ is continuous on $\triangle_K \times \mathcal{M}$,*

2. *$\boldsymbol{\mu} \mapsto D(\boldsymbol{\mu}, \neg i)$ and $\boldsymbol{\mu} \mapsto D(\boldsymbol{\mu})$ are continuous on $\mathcal{M}$,*

3. *$\boldsymbol{\mu} \mapsto \boldsymbol{w}^*(\boldsymbol{\mu}, \neg i)$, $\boldsymbol{\mu} \mapsto \boldsymbol{w}^*(\boldsymbol{\mu})$ and $\boldsymbol{\mu} \mapsto i_F(\boldsymbol{\mu})$ are upper hemicontinuous on $\mathcal{M}$ with non-empty and compact values,*

The proof is in Appendix F. It uses Berge's maximum theorem and a modification thereof due to [Feinberg et al., 2014]. Related continuity results using this type of arguments, but restricted to single-valued functions, appeared for the regret minimization problem in [Combes et al., 2017].

## 4.2 Convexity

Next we establish convexity.

**Proposition 5.** *For each $i \in \mathcal{I}$, for all $\boldsymbol{\mu} \in \mathcal{M}$ the set $\boldsymbol{w}^*(\boldsymbol{\mu}, \neg i)$ is convex.*

This is a consequence of the concavity of $\boldsymbol{w} \mapsto D(\boldsymbol{w}, \boldsymbol{\mu}, \neg i)$ (which is an infimum of linear functions). In single-answer problems, we obtain that the oracle weights set $\boldsymbol{w}^*(\boldsymbol{\mu})$ is convex everywhere. This is however not the case in general for multiple-answer problems, as illustrated by the next example.

**Example 1** (Continued). Consider a $K = 2$-arm *Any Low Arm* instance $\boldsymbol{\mu}$ with $\mu_1 < \gamma$ and $\mu_2 < \gamma$, so that both answers 1 and 2 are correct. Recall that $D(\boldsymbol{\mu}) = \max_{k=1,2} d(\mu_k, \gamma)$. Now for $\mu_1 < \mu_2 < \gamma$, $\boldsymbol{w}^*(\boldsymbol{\mu}) = \{(1, 0)\}$ and symmetrically for $\mu_2 < \mu_1 < \gamma$, $\boldsymbol{w}^*(\boldsymbol{\mu}) = \{(0, 1)\}$. However, for $\mu_1 = \mu_2 < \gamma$, $\boldsymbol{w}^*(\boldsymbol{\mu}) = \{(1, 0), (0, 1)\}$, which is not convex. Playing intermediate weights $\boldsymbol{w} = (\alpha, 1 - \alpha)$ results in strictly sub-optimal $D(\boldsymbol{\mu}, \boldsymbol{w}) = \max\{\alpha, 1 - \alpha\} d(\mu, \gamma) < d(\mu, \gamma) = D(\boldsymbol{\mu})$.

This example also illustrates the upper hemicontinuity of $\boldsymbol{w}^*(\boldsymbol{\mu})$: since $\boldsymbol{\mu}$ of the form $(\mu, \mu)$ is the limit of a sequence $(\boldsymbol{\mu}_t)_{t \in \mathbb{N}}$ with $\mu_{t,1} < \mu_{t,2}$, we obtain that $\{(1, 0)\} \subseteq \boldsymbol{w}^*(\boldsymbol{\mu})$. Similarly, using a sequence with $\mu_{t,1} > \mu_{t,2}$, $\{(0, 1)\} \subseteq \boldsymbol{w}^*(\boldsymbol{\mu})$.

The example scales up to $K$ arms, and shows that the sample complexity guarantee for vanilla TaS (Theorem 9) may exceeds by a factor $K$ the optimal complexity, which is matched by our new method (Theorem 11).

## 4.3 Consequences for Track-and-Stop

The original analysis of Track-and-Stop excludes the mean vectors $\boldsymbol{\mu} \in \mathcal{M}$ for which $\boldsymbol{w}^*(\boldsymbol{\mu})$ is not a singleton. We show that the upper hemicontinuity and convexity properties of $\boldsymbol{w}^*(\boldsymbol{\mu})$ allow us to extend that analysis to all $\boldsymbol{\mu}$ with a single oracle answer (in particular all single-answer bandit problems), at least for one of the two Track-and-Stop variants. Indeed, that algorithm was introduced with two possible subroutines, dubbed C-tracking and D-tracking [Garivier and Kaufmann, 2016]. Both variants compute oracle weights $\boldsymbol{w}_t$ at the point $\hat{\boldsymbol{\mu}}_t$, but the arm pulled differs.

*C-tracking*: compute the projection $\boldsymbol{w}_t^{\varepsilon_t}$ of $\boldsymbol{w}_t$ on $\triangle_K^{\varepsilon_t} = \{\boldsymbol{w} \in \triangle_K : \forall k \in [K], w_k \geq \varepsilon_t\}$, where $\varepsilon_t > 0$. Pull the arm with index $k_t = \arg\min_{k \in [K]} N_{t,k} - \sum_{s=1}^t w_{s,k}^{\varepsilon_s}$.

*D-tracking*: if there is an arm $j$ with $N_{t,j} \leq \sqrt{t} - K/2$, then pull $k_t = j$. Otherwise, pull the arm $k_t = \arg\min_{k \in [K]} N_{t,k} - t w_{t,k}$.

The proof of the optimal sample complexity of Track-and-Stop for C-tracking remains essentially unchanged but we replace Proposition 9 of [Garivier and Kaufmann, 2016] by the following lemma, proved in Appendix G.3.

**Lemma 6.** *Let a sequence $(\hat{\boldsymbol{\mu}}_t)_{t \in \mathbb{N}}$ verify $\lim_{t \to +\infty} \hat{\boldsymbol{\mu}}_t = \boldsymbol{\mu}$. For all $t \geq 0$, let $\boldsymbol{w}_t \in \boldsymbol{w}^*(\hat{\boldsymbol{\mu}}_t)$ be arbitrary oracle weights for $\hat{\boldsymbol{\mu}}_t$. If $\boldsymbol{w}^*(\boldsymbol{\mu})$ is convex, then*

$$\lim_{t \to +\infty} \inf_{\boldsymbol{w} \in \boldsymbol{w}^*(\boldsymbol{\mu})} \left\| \frac{1}{t} \sum_{s=1}^t \boldsymbol{w}_s - \boldsymbol{w} \right\|_\infty = 0 .$$

The average of oracle weights for $\hat{\boldsymbol{\mu}}_t$ converges to the set of oracle weights for $\boldsymbol{\mu}$. C-tracking then ensures that the proportion of pulls $N_t/t$ is close to that average by Lemma 7 of [Garivier and Kaufmann, 2016], hence $N_t/t$ gets close to oracle weights.

**Theorem 7.** *For all $\boldsymbol{\mu} \in \mathcal{M}$ such that $i_F(\boldsymbol{\mu})$ is a singleton (in particular all single-answer problems), Track-and-Stop with C-tracking is $\delta$-correct with asymptotically optimal sample complexity.*

Proof in Appendix G.6. We encourage the reader to first proceed to Section 5, since the proof considers the result as a special case of the multiple-answers setting.

**Remark 8.** *If $\boldsymbol{w}^*(\boldsymbol{\mu})$ is not a singleton, Track-and-Stop using D-tracking may fail to converge to $\boldsymbol{w}^*(\boldsymbol{\mu})$, even when it is convex.*

While we do not prove that D-tracking fails to converge to $\boldsymbol{w}^*(\boldsymbol{\mu})$ on a specific example of a bandit, we provide empirical evidence in Appendix E. The reason for the failure of D-tracking

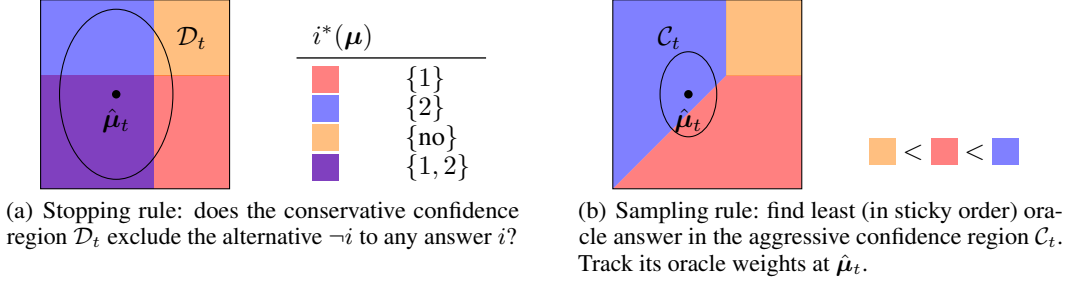

(a) Stopping rule: does the conservative confidence region $\mathcal{D}_t$ exclude the alternative $\neg i$ to any answer $i$?

(b) Sampling rule: find least (in sticky order) oracle answer in the aggressive confidence region $\mathcal{C}_t$. Track its oracle weights at $\hat{\boldsymbol{\mu}}_t$.

Figure 1: Sticky Track-and-Stop: The two main ideas, illustrated on the *Any Low Arm* problem.

is that it does not in general converge to the convex hull of the points it tracks. Suppose that $\boldsymbol{w}_t = \boldsymbol{w}^{(1)} = (1/2, 1/2, 0)$ for $t$ odd and $\boldsymbol{w}_t = \boldsymbol{w}^{(2)} = (1/2, 0, 1/2)$ for $t$ even. Then D-tracking verifies $\lim_{t \to +\infty} N_t/t = (1/3, 1/3, 1/3)$. This limit is outside of the convex hull of $\{\boldsymbol{w}^{(1)}, \boldsymbol{w}^{(2)}\}$.

## 5 Algorithms for the Multiple-Answers Setting

We can prove for Track-and-Stop the following suboptimal upper bound on the sample complexity, based on the fact that it ensures convergence of $N_t/t$ to the convex hull of the oracle weight set.

**Theorem 9.** *Let* $\operatorname{conv}(A)$ *be the convex hull of a set $A$. For all $\boldsymbol{\mu} \in \mathcal{M}$ in a multi-answer problem, Track-and-Stop with C-tracking is $\delta$-correct and verifies*

$$\lim_{\delta \to 0} \frac{\mathbb{E}_{\boldsymbol{\mu}}[\tau_\delta]}{\log(1/\delta)} \leq \max_{\boldsymbol{w} \in \operatorname{conv}(\boldsymbol{w}^*(\boldsymbol{\mu}))} \frac{1}{D(\boldsymbol{w}, \boldsymbol{\mu})} \ .$$

### 5.1 Sticky Track-and-Stop

The cases of multiple-answers problems for which Track-and-Stop is inadequate are $\boldsymbol{\mu} \in \mathcal{M}$ with $i_F(\boldsymbol{\mu})$ of cardinality greater than 1. When convexity does not hold, $\boldsymbol{w}^*(\boldsymbol{\mu})$ is the union of the convex sets $(\boldsymbol{w}^*(\boldsymbol{\mu}, \neg i))_{i \in i_F(\boldsymbol{\mu})}$. If an algorithm can a priori select $i_f \in i_F(\boldsymbol{\mu})$ and track allocations $\boldsymbol{w}_t$ in $\boldsymbol{w}^*(\hat{\boldsymbol{\mu}}_t, \neg i_f)$, then using Track-and-Stop on that restricted problem will ensure that $N_t/t$ converges to the oracle weights. Our proposed algorithm, Sticky Track-and-Stop, which we display in Algorithm 1, uses a confidence region around the current estimate $\hat{\boldsymbol{\mu}}_t$ to determine what $i \in \mathcal{I}$ can be the oracle answer for $\boldsymbol{\mu}$. It selects one of these answers according to an arbitrary total order on $\mathcal{I}$ and does not change it (sticks to it) until no point in the confidence region has the chosen answer in its set of oracle answers.

---

**Algorithm 1** Sticky Track-and-Stop.

---

Input: $\delta > 0$, strict total order on $\mathcal{I}$. Set $t = 1$, $\hat{\boldsymbol{\mu}}_0 = 0$, $N_0 = 0$.
**while** *not stopped* **do**
    Let $\mathcal{C}_t = \{\boldsymbol{\mu}' \in \mathcal{M} : D(N_{t-1}, \hat{\boldsymbol{\mu}}_{t-1}, \boldsymbol{\mu}') \leq \log(f(t-1))\}$ .       `// small conf. reg.`
    Compute $I_t = \bigcup_{\boldsymbol{\mu}' \in \mathcal{C}_t} i_F(\boldsymbol{\mu}')$ .
    Pick the first alternative $i_t \in I_t$ in the order on $\mathcal{I}$.
    Compute $\boldsymbol{w}_t \in \boldsymbol{w}^*(\hat{\boldsymbol{\mu}}_{t-1}, \neg i_t)$.
    Pull an arm $a_t$ according to the C-tracking rule and receive $X_t \sim \nu_{a_t}$ .
    Set $N_t = N_{t-1} + \boldsymbol{e}_{a_t}$ and $\hat{\boldsymbol{\mu}}_t = \hat{\boldsymbol{\mu}}_{t-1} + \frac{1}{N_{t,a_t}}(X_t - \hat{\mu}_{t-1,a_t})\boldsymbol{e}_{a_t}$ .
    Let $\mathcal{D}_t = \{\boldsymbol{\mu}' \in \mathcal{M} : D(N_t, \hat{\boldsymbol{\mu}}_t, \boldsymbol{\mu}') \leq \beta(t, \delta)\}$ .          `// large conf. reg.`
    **if** *there exists $i \in \mathcal{I}$ such that $\mathcal{D}_t \cap \neg i = \emptyset$* **then**
        stop and return $i$.
    **end**
    $t \leftarrow t + 1$ .
**end**

---

**Theorem 10.** *For $\beta(t, \delta) = \log(Ct^2/\delta)$, with $C$ such that $C \geq e \sum_{t=1}^{+\infty} (\frac{e}{K})^K \frac{(\log^2(Ct^2)\log(t))^K}{t^2}$, Sticky Track-and-Stop is $\delta$-correct.*

That result is a consequence of Proposition 12 of [Garivier and Kaufmann, 2016].

## 5.2 Sample Complexity

**Theorem 11.** *Sticky Track-and-Stop is asymptotically optimal, i.e. it verifies for all $\boldsymbol{\mu} \in \mathcal{M}$,*

$$\lim_{\delta \to 0} \frac{\mathbb{E}_{\boldsymbol{\mu}}[\tau_\delta]}{\log(1/\delta)} \to \frac{1}{D(\boldsymbol{\mu})} \, .$$

Let $i_{\boldsymbol{\mu}} = \min i_F(\boldsymbol{\mu})$ in the arbitrary order on answers. For $\varepsilon, \xi > 0$, we define $C^*_{\varepsilon,\xi}(\boldsymbol{\mu})$, the minimal value of $D(\boldsymbol{w}', \boldsymbol{\mu}', \neg i_{\boldsymbol{\mu}})$ for $\boldsymbol{w}'$ and $\boldsymbol{\mu}'$ in $\varepsilon$ and $\xi$-neighbourhoods of $\boldsymbol{w}^*(\boldsymbol{\mu})$ and $\boldsymbol{\mu}$.

$$C^*_{\varepsilon,\xi}(\boldsymbol{\mu}) = \inf_{\substack{\boldsymbol{\mu}':\|\boldsymbol{\mu}'-\boldsymbol{\mu}\|_\infty \leq \xi \\ \boldsymbol{w}':\inf_{\boldsymbol{w} \in \boldsymbol{w}^*(\boldsymbol{\mu}, \neg i_{\boldsymbol{\mu}})} \|\boldsymbol{w}'-\boldsymbol{w}\|_\infty \leq 3\varepsilon}} D(\boldsymbol{w}', \boldsymbol{\mu}', \neg i_{\boldsymbol{\mu}}) \, .$$

Our proof strategy is to show that under a concentration event defined below, for $t$ big enough, $(\hat{\boldsymbol{\mu}}_t, N_t/t)$ belongs to that $(\xi, \varepsilon)$ neighbourhood of $(\boldsymbol{\mu}, \boldsymbol{w}^*(\boldsymbol{\mu}, \neg i_{\boldsymbol{\mu}}))$. From that fact, we obtain $D(N_t, \hat{\boldsymbol{\mu}}_t, \neg i_{\boldsymbol{\mu}}) \geq t C^*_{\varepsilon,\xi}(\boldsymbol{\mu})$. Furthermore, if the algorithm does not stop at stage $t$, we also get an upper bound on $D(N_t, \hat{\boldsymbol{\mu}}_t, \neg i_{\boldsymbol{\mu}})$ from the stopping condition. We obtain an upper bound on the stopping time, function of $\delta$ and $C^*_{\varepsilon,\xi}(\boldsymbol{\mu})$. By continuity of $(\boldsymbol{w}, \boldsymbol{\mu}) \mapsto D(\boldsymbol{w}, \boldsymbol{\mu}, \neg i_{\boldsymbol{\mu}})$ (from Theorem 4), we have $\lim_{\varepsilon \to 0, \xi \to 0} C^*_{\varepsilon,\xi}(\boldsymbol{\mu}) = D(\boldsymbol{\mu}, \neg i_{\boldsymbol{\mu}}) = D(\boldsymbol{\mu})$.

**Two concentration events.** Let $\mathcal{E}_T = \bigcap_{t=h(T)}^{T} \{\boldsymbol{\mu} \in \mathcal{C}_t\}$ be the event that the small confidence region contains the true parameter vector $\boldsymbol{\mu}$ for $t \geq h(T)$. The function $h : \mathbb{N} \to \mathbb{R}$, positive, increasing and going to $+\infty$, makes sure that each event $\{\boldsymbol{\mu} \in \mathcal{C}_t\}$ appears in finitely many $\mathcal{E}_T$, which will be essential in the concentration results. We will eventually use $h(T) = \sqrt{T}$.

In order to define the second event, we first highlight a consequence of Theorem 4.

**Corollary 12.** *For all $\varepsilon > 0$, for all $\boldsymbol{\mu} \in \mathcal{M}$, for all $i \in \mathcal{I}$, there exists $\xi > 0$ such that*

$$\|\boldsymbol{\mu}' - \boldsymbol{\mu}\|_\infty \leq \xi \Rightarrow \forall \boldsymbol{w}' \in \boldsymbol{w}^*(\boldsymbol{\mu}', \neg i) \ \exists \boldsymbol{w} \in \boldsymbol{w}^*(\boldsymbol{\mu}, \neg i), \ \|\boldsymbol{w}' - \boldsymbol{w}\|_\infty \leq \varepsilon \, .$$

Let $\mathcal{E}'_T = \bigcap_{t=h(T)}^{T} \{\|\hat{\boldsymbol{\mu}}_t - \boldsymbol{\mu}\|_\infty \leq \xi\}$ be the event that the empirical parameter vector is close to $\boldsymbol{\mu}$, where $\xi$ is chosen as in the previous corollary for $i = i_{\boldsymbol{\mu}}$. The analysis of Sticky Track-and-Stop consists of two parts: first show that $\mathcal{E}_T^c$ and $\mathcal{E}'_T{}^c$ happen rarely enough to lead only to a finite term in $\mathbb{E}_{\boldsymbol{\mu}}[\tau_\delta]$; then show than under $\mathcal{E}_T \cap \mathcal{E}'_T$ there is an upper bound on $\tau_\delta$.

**Lemma 13.** *Suppose that there exists $T_0$ such that for $T \geq T_0$, $\mathcal{E}_T \cap \mathcal{E}'_T \subset \{\tau_\delta \leq T\}$. Then*

$$\mathbb{E}_{\boldsymbol{\mu}}[\tau_\delta] \leq T_0 + \sum_{T=T_0}^{+\infty} \mathbb{P}_{\boldsymbol{\mu}}(\mathcal{E}_T^c) + \sum_{T=T_0}^{+\infty} \mathbb{P}_{\boldsymbol{\mu}}(\mathcal{E}'_T{}^c) \, . \tag{1}$$

*Proof.* Since $\tau_\delta$ is a non-negative integer-valued random variable, $\mathbb{E}_{\boldsymbol{\mu}}[\tau_\delta] = \sum_{T=0}^{+\infty} \mathbb{P}_{\boldsymbol{\mu}}\{\tau_\delta > T\}$. For $T \geq T_0$, $\mathbb{P}_{\boldsymbol{\mu}}\{\tau_\delta > T\} \leq \mathbb{P}_{\boldsymbol{\mu}}(\mathcal{E}_T^c \cup \mathcal{E}'_T{}^c) \leq \mathbb{P}_{\boldsymbol{\mu}}(\mathcal{E}_T^c) + \mathbb{P}_{\boldsymbol{\mu}}(\mathcal{E}'_T{}^c)$. □

The sums depending on the events $\mathcal{E}_T$ and $\mathcal{E}'_T$ in (1) are finite for well chosen $h(T)$ and $\mathcal{C}(t)$.

**Lemma 14.** *For $h(T) = \sqrt{T}$ and $f(t) = \exp(\beta(t, 1/t^5)) = Ct^{10}$ in the definition of the confidence region $\mathcal{C}_t$, the sum $\sum_{T=T_0}^{+\infty} \mathbb{P}_{\boldsymbol{\mu}}(\mathcal{E}_T^c) + \sum_{T=T_0}^{+\infty} \mathbb{P}_{\boldsymbol{\mu}}(\mathcal{E}'_T{}^c)$ is finite.*

The proof of the Lemma can be found in Appendix G.1. The remainder of the proof is concerned with finding a suitable $T_0$. First, we show that if $\hat{\boldsymbol{\mu}}_t$ and $N_t/t$ are in an $(\xi, \varepsilon)$ neighbourhood of $\boldsymbol{\mu}$ and $\boldsymbol{w}^*(\boldsymbol{\mu}, \neg i_{\boldsymbol{\mu}})$, then such an upper bound $T_0$ on $\tau_\delta$ can be obtained.

**Lemma 15.** *Let $t_1$ be an integer and suppose that for all $t \geq t_1$, $D(N_t, \hat{\boldsymbol{\mu}}_t, \neg i_{\boldsymbol{\mu}}) \geq t C^*_{\varepsilon,\xi}(\boldsymbol{\mu})$. Let $T_\beta = \inf\{t : t > \beta(t, \delta)/C^*_{\varepsilon,\xi}(\boldsymbol{\mu})\}$. Then $\tau_\delta \leq \max(t_1, T_\beta)$.*

*Proof.* Take $t \geq t_1$. If $\tau_\delta > t$ then by hypothesis and the stopping condition, $t \leq D(N_t, \hat{\boldsymbol{\mu}}_t, \neg i_{\boldsymbol{\mu}})/C^*_{\varepsilon,\xi}(\boldsymbol{\mu}) \leq \beta(t, \delta)/C^*_{\varepsilon,\xi}(\boldsymbol{\mu})$. Conversely, for $t \geq t_1$, if $t > \beta(t, \delta)/C^*_{\varepsilon,\xi}(\boldsymbol{\mu})$ then $\tau_\delta \leq t$. We obtain that $\tau_\delta \leq \max(t_1, \inf\{t : t > \beta(t, \delta)/C^*_{\varepsilon,\xi}(\boldsymbol{\mu})\})$. □

**The oracle answer $i_t$ becomes constant.** Due to the forced exploration present in the C-tracking procedure, the confidence region $\mathcal{C}_t$ shrinks. After some time, when concentration holds, the set of possible oracle answers $I_t$ becomes constant over $t$ and equal to $i_F(\boldsymbol{\mu})$.

**Lemma 16.** *If an algorithm guaranties that for all $k \in [K]$ and all $t \geq 1$, $N_{t,k} \geq n(t) > 0$ with $\lim_{t \to +\infty} n(t)/\log(f(t)) = +\infty$, then there exists $T_\Delta$ such that under the event $\mathcal{E}_T$, for $t \geq \max(h(T), T_\Delta)$, $I_t = i_F(\boldsymbol{\mu})$ and $\min I_t = i_{\boldsymbol{\mu}} = \min i_F(\boldsymbol{\mu})$.*

Proof in Appendix G.4. Note that Lemma 16 depends only on the amount of forced exploration and not on other details of the algorithm. Any algorithm using C-tracking verifies the hypothesis with $n(t) = \sqrt{t + K^2} - 2K$ by Lemma 34 [Garivier and Kaufmann, 2016, Lemma 7].

**Convergence to the neighbourhood of $(\boldsymbol{\mu}, \boldsymbol{w}^*(\boldsymbol{\mu}, \neg i_{\boldsymbol{\mu}}))$.** Once $i_t = i_{\boldsymbol{\mu}}$, we fall back to tracking points from a convex set of oracle weights. The estimate $\hat{\boldsymbol{\mu}}_t$ and $N_t/t$ both converge, to $\boldsymbol{\mu}$ and to the set $\boldsymbol{w}^*(\boldsymbol{\mu}, \neg i_{\boldsymbol{\mu}})$. The Lemma below is proved in Appendix G.5.

**Lemma 17.** *Let $T_\Delta$ be defined as in Lemma 16. For $T$ such that $h(T) \geq T_\Delta$, it holds that on $\mathcal{E}_T \cap \mathcal{E}'_T$ Sticky Track-and-Stop with C-Tracking verifies*

$$\forall t \geq h(T), \|\hat{\boldsymbol{\mu}}_t - \boldsymbol{\mu}\|_\infty \leq \xi, \quad and \quad \forall t \geq 4\frac{K^2}{\varepsilon^2} + 3\frac{h(T)}{\varepsilon}, \quad \inf_{\boldsymbol{w} \in \boldsymbol{w}^*(\boldsymbol{\mu}, \neg i_{\boldsymbol{\mu}})} \|\frac{\boldsymbol{N}_t}{t} - \boldsymbol{w}\|_\infty \leq 3\varepsilon.$$

**Remainder of the proof.** Suppose that the event $\mathcal{E}_T \cap \mathcal{E}'_T$ holds. Let $T_\Delta$ be defined as in Lemma 16 and $T$ be such that $h(T) \geq T_\Delta$. Let $\eta(T) = 4K^2/\varepsilon^2 + 3h(T)/\varepsilon$. For all $t \geq \eta(T)$ we have $D(N_t, \hat{\boldsymbol{\mu}}_t, \neg i_{\boldsymbol{\mu}}) \geq tC^*_{\varepsilon, \xi}(\boldsymbol{\mu})$ by Lemma 17. For $h(T)$ bigger than some $T_\eta$ we have $\eta(T) \leq T$. We suppose $h(T) \geq \max(T_\Delta, T_\eta)$. We apply Lemma 15 with $t_1 = \eta(T)$. We obtain that $\tau_\delta \leq \max(\eta(T), T_\beta) \leq \max(T, T_\beta)$. Conclusion: for $T \geq T_0 = \max(h^{-1}(T_\Delta), h^{-1}(T_\eta), T_\beta)$, under the concentration event, $\tau_\delta \leq T$ and we can apply Lemma 13.

Note that $\lim_{\delta \to 0} \frac{T_0}{\log(1/\delta)} = \frac{1}{C^*_{\varepsilon,\xi}(\boldsymbol{\mu})}$. Taking $\varepsilon \to 0$ (hence $\xi \to 0$ as well), we obtain $\lim_{\delta \to 0} \frac{\mathbb{E}_{\boldsymbol{\mu}}[\tau_\delta]}{\log(1/\delta)} = \frac{1}{\lim_{\varepsilon \to 0} C^*_{\varepsilon,\xi}(\boldsymbol{\mu})} = \frac{1}{D(\boldsymbol{\mu})}$. We proved Theorem 11.

## 6 Conclusion

We characterized the complexity of multiple-answers pure exploration bandit problems, showing a lower bound and exhibiting an algorithm with asymptotically matching sample complexity on all such problems. That study could be extended in several interesting directions and we now list a few.

• The computational complexity of Track-and-Stop is an important issue: it would be desirable to design a pure exploration algorithm with optimal sample complexity which does not need to solve a min-max problem at each step. Furthermore, the same would need to be done for the sticky selection of an answer for the multiple-answers setting.
• Both lower bounds and upper bounds in this paper are asymptotic. In the upper bound case, only the forced exploration rounds are considered when evaluating the convergence of $\hat{\boldsymbol{\mu}}_t$ to $\boldsymbol{\mu}$, giving rise to potentially sub-optimal lower order terms. A finite time analysis with reasonably small $o(\log(1/\delta))$ terms for an optimal algorithm is desirable. In addition, while selecting one of the oracle answers to stick to has no asymptotic cost, it could have a lower order effect on the sample complexity and appear in a refined lower bound.
• Current tools in the theory of Brownian motion are insufficient to characterise the asymptotic distribution of proportions induced by tracking, even for two arms. Without tracking the Arcsine law arises, so this slightly more challenging problem holds the promise of similarly elegant results.
• Finally, the multiple answer pure exploration setting can be extended in various ways. Making $\mathcal{I}$ continuous leads to regression problems. The parametric assumption that the arms are in one-parameter exponential families could also be relaxed.

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
