[Supplementary Material]

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

# A Notations

| Concept | Symbol |
|---|---|
| Exponential family mean parameter | $\mu \in \mathcal{O}$ |
| Bandit model | $\boldsymbol{\mu} \in \mathcal{M} \subseteq \mathcal{O}^K$ |
| Possible answers | $\mathcal{I}$ |
| Correct answer for bandit model $\boldsymbol{\mu}$ | $i^*(\boldsymbol{\mu}) : \mathcal{M} \to 2^{\mathcal{I}}$ |
| Alternative to $i \in \mathcal{I}$ | $\neg i = \{\boldsymbol{\mu} \in \mathcal{M} : i \notin i^*(\boldsymbol{\mu})\}$ |
| $K$-Simplex | $\triangle_K$ |
| Non-negative orthant | $\mathcal{Q}_+$ |
| interior, closure, convex hull of set $A$ | $\mathring{A}$, $\mathrm{cl}(A)$, $\mathrm{conv}(A)$ |
| Oracle answers and weights | $i_F(\boldsymbol{\mu}) : \mathcal{M} \to 2^{\mathcal{I}}$, $\boldsymbol{w}^*(\boldsymbol{\mu}) : \mathcal{M} \to \mathcal{P}(\triangle_K)$ |
| Bandit arm | $k \in [K]$ |
| Number of samples of arm $k$ at time $t$ | $N_{t,k}$ |
| mean of samples of arm $k$ at time $t$ | $\hat{\mu}_{t,k}$ |

# B Visualisation of Multiple-Answer Pure Exploration Problems

| Identification Problem | Possible answers $\mathcal{I}$ | Correct answers $i^*(\boldsymbol{\mu}) \subseteq \mathcal{I}$ | | Correct $i^*(\boldsymbol{\mu})$ | Oracle $i_F(\boldsymbol{\mu})$ |
|---|---|---|---|---|---|
| $\epsilon$ Best Arm | $[K]$ | $\{k \mid \mu_k \geq \max_j \mu_j - \epsilon\}$ | |  |  |
| Thresholding Bandit | $2^K$ | $\{\{k\|\mu_k \leq \gamma\}\}$ | |  |  |
| $\epsilon$ Minimum Threshold | $\{\mathrm{lo}, \mathrm{hi}\}$ | $\{\mathrm{lo}\}$ $\{\mathrm{hi}\}$ $\{\mathrm{lo}, \mathrm{hi}\}$ | if $\min_k \mu_k < \gamma - \epsilon$ if $\min_k \mu_k > \gamma + \epsilon$ o.w. |  |  |
| Any Low Arm | $[K] \cup \{\mathrm{no}\}$ | $\{k\|\mu_k \leq \gamma\}$ $\{\mathrm{no}\}$ | if $\min_k \mu_k < \gamma$ if $\min_k \mu_k > \gamma$ |  |  |
| Any Sign | $[K] \times \{\mathrm{lo}, \mathrm{hi}\}$ | $\{(k, \mathrm{lo})\|\mu_k \leq \gamma\} \cup \{(k, \mathrm{hi})\|\mu_k \geq \gamma\}$ | |  |  |

Table 1: Collection of Identification Problems. The diagrams depict 2-arm instances, parameterised by the two means, with colours showing the set of correct answers: **one correct answer:** ▮ $\{1\}$, ▮ $\{2\}$, ▮ $\{\{1\}\}$, ▮ $\{\{2\}\}$, ▮ $\{\{1, 2\}\}$, ▮ $\{\mathrm{lo}\}$, ▮ $\{\emptyset\}/\{\mathrm{hi}\}/\{\mathrm{no}\}$, ▮ $\{(1, \mathrm{lo})\}$, ▮ $\{(1, \mathrm{hi})\}$, ▮ $\{(2, \mathrm{lo})\}$, ▮ $\{(2, \mathrm{hi})\}$, and **two correct answers:** ▮ $\{1, 2\}$, ▮ $\{\mathrm{lo}, \mathrm{hi}\}$, ▮ $\{(1, \mathrm{lo}), (2, \mathrm{lo})\}$, ▮ $\{(1, \mathrm{lo}), (2, \mathrm{hi})\}$, ▮ $\{(1, \mathrm{hi}), (2, \mathrm{lo})\}$, ▮ $\{(1, \mathrm{hi}), (2, \mathrm{hi})\}$ .

# C Lower bound Proofs

In this section we build up to the proof of the lower bound Theorem 1. We start with the minimax result Lemma 2.

## C.1 Lemma 2: Characteristic time as the value of a game

Let $\neg j$ be the set of bandit problems for which $j$ is not a valid answer. Let us define the characteristic time by

$$\frac{1}{T_j^*(\boldsymbol{\mu})} = D(\boldsymbol{\mu}, \neg j) = \sup_{\boldsymbol{w}} \inf_{\boldsymbol{\lambda} \in \neg j} \sum_k w_k d(\mu_k, \lambda_k).$$

We now view the characteristic time problem as defining a two-player zero-sum game, with the purpose of obtaining a minimax optimal mixed strategy for the $\inf$ player. We will use such a mixed strategy to construct hard learning problems for proving sample complexity lower bounds in the next section. In this section we focus on the existence of the minimax strategy. We define the *bandit complexity* game to be the semi-infinite two-player zero-sum simultaneous game where:

- MAX's pure strategies are arms $i \in [K] = \{1, \ldots, K\}$,
- MIN's pure strategies are bandit models $\boldsymbol{\lambda} \in \neg j \subseteq \mathcal{M}$ (we may equivalently have MIN play a point $s \in S := \left\{ \big( d(\mu_1, \lambda_1), \ldots, d(\mu_K, \lambda_K) \big) \big| \boldsymbol{\lambda} \in \neg j \right\} \subseteq [0, \infty)^K$),
- the payoff function is $(i, \boldsymbol{\lambda}) \mapsto d(\mu_i, \lambda_i)$ (or, equivalently, $(i, s) \mapsto s_i$).

By definition, $D(\boldsymbol{\mu}, \neg j)$ is the optimal payoff when MAX randomises and plays first. We aim to show that a matching randomised strategy exists for when MIN plays first. That is, we want to establish a min-max theorem.

*of Lemma 2.* Combining (a) a standard application of Sion's minimax theorem to the bilinear function $f : \triangle \times \text{conv}(S) \to \mathbb{R}$ defined by $f(\boldsymbol{w}, s) = \langle \boldsymbol{w}, s \rangle$ and (b) the support size insight of Blackwell and Girshick [1954, Theorem 2.4.2] yields the Lemma. $\qquad\square$

For convenience, we will assume in the remainder that the infimum above is attained (e.g. when the convex hull of $S$ is compact), possibly on the closure of $\neg j$. If it is not, we need to apply the below arguments to a sequence of $\epsilon$-suboptimal $\mathbb{P}$ and let $\epsilon \to 0$. At any rate, we assume there exist $\boldsymbol{\lambda}^1, \ldots, \boldsymbol{\lambda}^K \in \neg j$ (or its closure) and $\boldsymbol{q} \in \triangle_K$ such that

$$\forall i : \sum_k q_k d(\mu_i, \lambda_i^k) \; \leq \; D(\boldsymbol{\mu}, \neg j). \tag{2}$$

## C.2 Consequences of the Minimax result

In this section we build up lower bounds by relating the probability of any event between two or more bandit problem. We start with a useful change of measures observation, used in [Garivier and Kaufmann, 2019] to derive a lower bound on the sample complexity of $\varepsilon$-Best Arm Identification problems.

**Proposition 18.** *Consider two distributions $\mathbb{P}$ and $\mathbb{Q}$. Let us denote the log-likelihood ratio after $n$ rounds by $L_n = \ln \frac{d\mathbb{P}}{d\mathbb{Q}}$. Then for any measurable event $A \in \mathcal{F}_n$ and threshold $\gamma \in \mathbb{R}$,*

$$\mathbb{Q}(A) \; \geq \; e^{-\gamma} \left( \mathbb{P}(A) - \mathbb{P}\{L_n > \gamma\} \right). \tag{3}$$

*Proof.*

$$\mathbb{Q}(A) = \mathbb{E}_{\mathbb{P}}[\mathbb{I}_A e^{-L_n}] \geq \mathbb{E}_{\mathbb{P}}[\mathbb{I}_{A \cap \{L_n \leq \gamma\}} e^{-L_n}]$$
$$\geq \mathbb{P}(A \cap \{L_n \leq \gamma\}) e^{-\gamma} \geq e^{-\gamma} (\mathbb{P}(A) - \mathbb{P}\{L_n > \gamma\}).$$

$\qquad\square$

## C.3 Likelihood ratio Martingales

Next we investigate the specific form of the likelihood ratio between two bandit models. Fix bandit models $\boldsymbol{\mu}$ and $\boldsymbol{\lambda}$, and any sampling strategy. Then after $n$ rounds,

$$\ln \frac{d\mathbb{P}_{\boldsymbol{\mu}}}{d\mathbb{P}_{\boldsymbol{\lambda}}} \; = \; \sum_i N_{n,i} \, \text{KL}(\nu_{\mu_i,i}, \nu_{\lambda_i,i}) + M_n(\boldsymbol{\mu}, \boldsymbol{\lambda})$$

where $M_n(\boldsymbol{\mu}, \boldsymbol{\lambda})$ is a martingale. To see this, we write $\text{KL}(\nu_{\mu,i}, \nu_{\lambda,i}) = d(\mu, \lambda) = \phi(\mu) - \phi(\lambda) - (\mu - \lambda)\phi'(\lambda)$, where we write $\phi$ for the convex generator of the Bregman divergence $d(\cdot, \cdot)$. Then

$$\ln \frac{d\mathbb{P}_{\boldsymbol{\mu}}}{d\mathbb{P}_{\boldsymbol{\lambda}}} \; = \; \sum_i N_{n,i} \left( d(\hat{\mu}_{n,i}, \lambda_i) - d(\hat{\mu}_{n,i}, \mu_i) \right)$$
$$= \; \sum_i N_{n,i} \Big( d(\mu_i, \lambda_i) + \big( \phi'(\mu_i) - \phi'(\lambda_i) \big) \big( \hat{\mu}_{n,i} - \mu_i \big) \Big)$$

hence $M_n(\boldsymbol{\mu}, \boldsymbol{\lambda}) = \sum_i N_{n,i} \big( \phi'(\mu_i) - \phi'(\lambda_i) \big) \big( \hat{\mu}_{n,i} - \mu_i \big)$.

## C.4 Exploiting the Minimax Distribution

We now bound the probability of any event between $\boldsymbol{\mu}$ and the hard problems given by the minimax distribution.

**Lemma 19.** *Fix a bandit model $\boldsymbol{\mu}$ with sub-Gaussian arm distributions. Let $\boldsymbol{q}$ and $\boldsymbol{\lambda}^1, \ldots, \boldsymbol{\lambda}^K$ be a minimax witness from Lemma 2, and let us introduce the abbreviation $\alpha_i = \phi'(\mu_i) - \sum_k q_k \phi'(\lambda_i^k)$. Fix sample size $n$, and consider any event $A \in \mathcal{F}_n$. Then for any $\beta > 0$*

$$
\max_{k \in [K]} \mathbb{P}_{\boldsymbol{\lambda}^k}\{A\} \geq e^{-\frac{n}{T^*(\boldsymbol{\mu})} - \beta} \left( \mathbb{P}_{\boldsymbol{\mu}}\{A\} - \exp\left( \frac{-\beta^2}{2n \max_i \alpha_i^2} \right) \right).
$$

In words, if $A$ is likely under $\boldsymbol{\mu}$ then it must also be likely under at least one $\boldsymbol{\lambda}^k$ for sample sizes $n \ll T^*(\boldsymbol{\mu})$.

*Proof.* Let us form the (Bayesian) mixture distribution $\mathbb{P}_{\boldsymbol{q}} = \sum_k q_k \mathbb{P}_{\boldsymbol{\lambda}^k}$. We have

$$
L_n = -\ln \frac{d\mathbb{P}_{\boldsymbol{q}}}{d\mathbb{P}_{\boldsymbol{\mu}}} \leq \sum_k q_k \ln \frac{d\mathbb{P}_{\boldsymbol{\mu}}}{d\mathbb{P}_{\boldsymbol{\lambda}^k}}.
$$

It follows that for any $\gamma \in \mathbb{R}$ we have

$$
\{L_n > \gamma\} \subseteq \left\{ \sum_k q_k \sum_i N_{n,i} d(\mu_i, \lambda_i^k) + \sum_k q_k M_n(\boldsymbol{\mu}, \boldsymbol{\lambda}^k) > \gamma \right\}.
$$

Picking $\gamma = \frac{n}{T^*(\boldsymbol{\mu})} + \beta$, we find

$$
= \left\{ \sum_k q_k \sum_i N_{n,i} d(\mu_i, \lambda_i^k) + \sum_k q_k M_n(\boldsymbol{\mu}, \boldsymbol{\lambda}^k) > \frac{n}{T^*(\boldsymbol{\mu})} + \beta \right\}
$$

Since $(\boldsymbol{w}^*, \boldsymbol{q})$ is a Nash equilibrium of the game and $N_n/n$ is a mixed strategy for the first player, $\sum_k q_k \sum_i N_{n,i} d(\mu_i, \lambda_i^k) \leq n \sum_k q_k \sum_i w_i^* d(\mu_i, \lambda_i^k) = \frac{n}{T^*(\boldsymbol{\mu})}$, so we find

$$
\subseteq \left\{ \sum_k q_k M_n(\boldsymbol{\mu}, \boldsymbol{\lambda}^k) > \beta \right\}
$$

$$
= \left\{ \sum_k q_k \sum_i N_{n,i} (\phi'(\mu_i) - \phi'(\lambda_i^k)) (\hat{\mu}_{n,i} - \mu_i) > \beta \right\}
$$

$$
= \left\{ \sum_i N_{n,i} \alpha_i (\hat{\mu}_{n,i} - \mu_i) > \beta \right\} \tag{4}
$$

The above left-hand quantity is a martingale of length $n$. Using the sub-Gaussianity assumption, the Hoeffding-Azuma inequality gives

$$
\mathbb{P}_{\boldsymbol{\mu}}\{L_n > \gamma\} \leq \exp\left( \frac{-\beta^2}{2n \max_i \alpha_i^2} \right).
$$

Let $A$ be a $\mathcal{F}_n$-measurable event. Combination with the change of measure argument (3) with $\max_k \mathbb{P}_{\boldsymbol{\lambda}^k}\{A\} \geq \mathbb{P}_{\boldsymbol{q}}\{A\}$ gives the result. $\qquad\square$

The sub-Gaussian assumption of the Lemma can undoubtedly be relaxed. The crucial requirement is that the $n$-step martingale in (4) concentrates, and hence cannot be large w.h.p.

We are now ready for the proof of Theorem 1, in which we will carefully tune the time $n$ to which we apply the above result.

## C.5 Proof of Theorem 1

We will bound the expectation of the stopping time $\tau_\delta$ through Markov's inequality. For $T > 0$,

$$\mathbb{E}_{\boldsymbol{\mu}}[\tau_\delta] \geq T(1 - \mathbb{P}_{\boldsymbol{\mu}}(\tau_\delta \leq T)) .$$

The event $\{\tau_\delta \leq T\}$ can be partitioned depending on the answer which is returned. Since the algorithm is $\delta$-PAC by hypothesis, $\mathbb{P}_{\boldsymbol{\mu}}(\tau_\delta \leq T, \hat{\imath}_\delta \notin i^*(\boldsymbol{\mu})) \leq \delta$. So

$$\mathbb{P}_{\boldsymbol{\mu}}(\tau_\delta \leq T) = \sum_i \mathbb{P}_{\boldsymbol{\mu}}(\tau_\delta \leq T, \hat{\imath}_\delta = i)$$

$$\leq \delta + \sum_{i \in i^*(\boldsymbol{\mu})} \mathbb{P}_{\boldsymbol{\mu}}(\tau_\delta \leq T, \hat{\imath}_\delta = i) .$$

For $i \in i^*(\boldsymbol{\mu})$, fix a minimax strategy $\boldsymbol{\lambda}^1, \dots, \boldsymbol{\lambda}^K \in \neg i$ and $\boldsymbol{q} \in \triangle_K$ as given by Lemma 2. Then by Lemma 19, for any $\beta > 0$

$$\mathbb{P}_{\boldsymbol{\mu}}(\tau_\delta \leq T, \hat{\imath}_\delta = i) \leq \exp\left(\frac{T}{T_i^*(\boldsymbol{\mu})} + \beta\right) \max_k \mathbb{P}_{\boldsymbol{\lambda}^k}(\tau_\delta \leq T, \hat{\imath}_\delta = i) + \exp\left(\frac{-\beta^2}{2T \max_i \alpha_i^2}\right)$$

$$\leq \delta \exp\left(\frac{T}{T_i^*(\boldsymbol{\mu})} + \beta\right) + \exp\left(\frac{-\beta^2}{2T \max_i \alpha_i^2}\right)$$

Let $\alpha^2 = \max_i \alpha_i^2$. For $\eta \in (0,1)$, $T \leq (1-\eta)T_i^*(\boldsymbol{\mu})\log(1/\delta)$ and $\beta = \frac{\eta}{2\sqrt{1-\eta}}\sqrt{\frac{T}{T_i^*(\boldsymbol{\mu})}\log(1/\delta)}$,

$$\mathbb{P}_{\boldsymbol{\mu}}(\tau_\delta \leq T, \hat{\imath}_\delta = i) \leq \delta \exp\left(\frac{T}{T_i^*(\boldsymbol{\mu})} + \frac{\eta}{2\sqrt{1-\eta}}\sqrt{\frac{T}{T_i^*(\boldsymbol{\mu})}\log(1/\delta)}\right) + \exp\left(\frac{-\eta^2 \log(1/\delta)}{8(1-\eta)T_i^*(\boldsymbol{\mu})\alpha^2}\right)$$

$$\leq \delta \exp\left((1-\eta/2)\log\frac{1}{\delta}\right) + \exp\left(\frac{-\eta^2 \log(1/\delta)}{8(1-\eta)T_i^*(\boldsymbol{\mu})\alpha^2}\right)$$

$$= \delta^{\eta/2} + \delta^{\eta^2/(8(1-\eta)T_i^*(\boldsymbol{\mu})\alpha^2)}$$

$$\xrightarrow[\delta \to 0]{} 0$$

Suppose that $T = \min_{i \in i^*(\boldsymbol{\mu})}(1-\eta)T_i^*(\boldsymbol{\mu})\log(1/\delta)$ for some $\eta \in (0,1)$. Then we must have $\lim_{\delta \to 0}\mathbb{P}_{\boldsymbol{\mu}}(\tau_\delta \leq T) = 0$ and therefore $\liminf_{\delta \to 0}\frac{\mathbb{E}_{\boldsymbol{\mu}}[\tau_\delta]}{\log(1/\delta)} \geq \lim_{\delta \to 0}\frac{T}{\log(1/\delta)}(1 - \mathbb{P}_{\boldsymbol{\mu}}(\tau_\delta \leq T)) = (1-\eta)\min_{i \in i^*(\boldsymbol{\mu})}T_i^*(\boldsymbol{\mu})$. Letting $\eta$ go to zero, we obtain that

$$\liminf_{\delta \to 0}\frac{\mathbb{E}_{\boldsymbol{\mu}}[\tau_\delta]}{\log(1/\delta)} \geq \min_{i \in i^*(\boldsymbol{\mu})} T_i^*(\boldsymbol{\mu}) .$$

# D  Vanilla Track and Stop Fails for Multiple Answers

We argue that Track and Stop in general does not ensure convergence of $N_t/t$ to $\boldsymbol{w}^*(\boldsymbol{\mu})$ when that set is not convex. We illustrate our claim on the *Any Half-Space* problem, a generalisation of *Any Sign* from Table 1. Given $n \in \mathbb{N}$ hyperplanes of $\mathbb{R}^K$ passing through 0, parametrized for $m \in [n]$ by normal vectors $\boldsymbol{u}_m \in \mathbb{R}^K$ with $\|\boldsymbol{u}_m\|_1 = 1$, the algorithm has to return $(m, s) \in [n] \times \{-1, 1\}$ such that $s\boldsymbol{\mu}^\intercal \boldsymbol{u}_m \geq 0$. i.e. it must return any of the half-spaces in which $\boldsymbol{\mu}$ lies. See Figure 3. The arms have Gaussian distributions with variance 1.

Figure 3: *Any Half-Space* problem. The oracle answers $i_F(\boldsymbol{\mu})$ are ▨ $\{\boldsymbol{u}_1\}$ and ▨ $\{\boldsymbol{u}_2\}$ (and both on the diagonal).

This problem is chosen for the simplicity of its $\boldsymbol{w}^*$ mapping. Indeed $\boldsymbol{w}^*(\boldsymbol{\mu}, \neg(m, s)) = \{\boldsymbol{u}_m\}$ if $s\boldsymbol{\mu}^\intercal \boldsymbol{u}_m \geq 0$ and $\boldsymbol{w}^*(\boldsymbol{\mu}, \neg(m, s)) = \triangle_K$ otherwise. The distance to that alternative is $D(\boldsymbol{\mu}, \neg(m, s)) = \mathbb{I}\{s\boldsymbol{\mu}^\intercal \boldsymbol{u}_m \geq 0\}(\boldsymbol{\mu}^\intercal \boldsymbol{u}_m)^2$. The optimal weights set $\boldsymbol{w}^*(\boldsymbol{\mu})$ is the union of those $\{\boldsymbol{u}_m\}$ for which the distance is the greatest, and can be non-convex.

(a) Histogram of stopping time $\tau$

(b) Distance of $\boldsymbol{N}_\tau/\tau$ to $\boldsymbol{w}^*(\boldsymbol{\mu})$ for TaS

Figure 2: Suboptimality of Track-and-Stop with $C$-tracking on a $K = 10$ arm instance of the *Any Half-Space* problem, where each $\mu_k = -1/10$. We run the algorithm with an excessively small $\delta = e^{-80}$ to focus on the asymptotic regime, using $500K$ repetitions. *Left*: empirical distribution of the stopping time of Track-and-Stop and of Sticky Track-and-Stop, with respective means 24K and 16K. *Right*: the reason for the suboptimality is that the sampling proportions of Track-and-Stop do not converge to the oracle weights.

For $a \in [0, 1]$, let $K = 2$, $n = 2$, $\boldsymbol{u}_1 = (a, 1 - a)$, $\boldsymbol{u}_2 = (1 - a, a)$ and $\boldsymbol{\mu} = (\mu_0, \mu_0)$ for some $\mu_0 \in \mathbb{R}$. Suppose that after stage $t_0$ of Track and Stop, $\hat{\boldsymbol{\mu}}_t$ verifies that $\hat{\boldsymbol{\mu}}_t^\mathsf{T} \boldsymbol{u}_m$ has same sign as $\boldsymbol{\mu}^\mathsf{T} \boldsymbol{u}_m$ for both $m \in \{1, 2\}$ (in expectation this happens except on at most a finite number of stages). Then $\boldsymbol{w}^*(\hat{\boldsymbol{\mu}}_t) = \{\boldsymbol{u}_1\}$ iff $\hat{\mu}_{t,1} > \hat{\mu}_{t,2}$ and $\boldsymbol{w}^*(\hat{\boldsymbol{\mu}}_t) = \{\boldsymbol{u}_2\}$ iff $\hat{\mu}_{t,1} < \hat{\mu}_{t,2}$. The case $\hat{\mu}_{t,1} = \hat{\mu}_{t,2}$ has probability 0, hence we ignore it.

C-tracking ensures that $N_t/t$ is close to $\sum_{s=1}^t \boldsymbol{w}_s$ for $\boldsymbol{w}_s \in \boldsymbol{w}^*(\hat{\boldsymbol{\mu}}_s)$. Calling $T_1(t)$ the number of stages up to $t$ for which $\hat{\mu}_{t,1} > \hat{\mu}_{t,2}$ and neglecting the first $t_0$ stages, $N_t/t \approx T_1(t)\boldsymbol{u}_1 + (1 - T_1(t))\boldsymbol{u}_2$. In the nomenclature of random walks, $T_1(t)$ is the occupation time of the region below the diagonal on Figure 3.

In order for Track and Stop to be optimal, $N_t/t$ need to be close to $\boldsymbol{w}^*(\boldsymbol{\mu}) = \{\boldsymbol{u}_1, \boldsymbol{u}_2\}$ at $t_{opt} = \log(1/\delta)/D(\boldsymbol{\mu})$. For $\boldsymbol{u}_1 \neq \boldsymbol{u}_2$ this means that the distribution of $T_1(t_{opt})/t_{opt}$ must be concentrated on $\{0, 1\}$. If the limit distribution of $T_1(t)/t$ (assuming it exists) for $t \to +\infty$ has mass in $(0, 1)$, Track and Stop likely has suboptimal asymptotic sample complexity.

In the case of $a = 1/2$, $\boldsymbol{u}_1 = \boldsymbol{u}_2 = (1/2, 1/2)$, fot $t$ even $N_{t,1} = N_{t,2}$ and $T_1(t) = \#\{s \le t : \sum_{u=1}^{s/2}(X_{2u}^{(1)} - X_{2u+1}^{(2)}) > 0\}$ is the occupation time of $\mathbb{R}^+$ for a Gaussian random walk. Its limit distribution is the Arcsine distribution. But in that case $N_t/t$ is always optimal. Experimentally, when $a \neq 1/2$ (hence $\boldsymbol{u}_1 \neq \boldsymbol{u}_2$ and $N_t/t$ not always optimal), we observe that the limit distribution for $T_1(t)/t$ is not Arcsine, but has mass in $(0, 1)$ for $a \in (0, 1)$. See Figure 4.

Figure 2 displays the stopping time of Track and Stop on such an hyperplane problem for $K = n = 10$ and shows that Track and Stop is empirically suboptimal and that $N_\tau/\tau$, proportions of pulls at the stopping time, is not concentrated near $\boldsymbol{w}^*(\boldsymbol{\mu})$.

## E  Failure of D-tracking

We illustrate the suboptimality of Track-and-Stop with D-tracking in a single-answer problem. On a 3-arms problem with Gaussian distributions with same variance, the algorithm must answer the two following queries:

- What is the sign of $\mu_1$?
- Is there $m \in \{1, 2\}$ such that $\boldsymbol{\mu}^\mathsf{T}\boldsymbol{u}^{(m)} \ge 0$? For some $a \in (0, 1)$, $\boldsymbol{u}^{(1)} = (0, a, 1 - a)$ and $\boldsymbol{u}^{(2)} = (0, 1 - a, a)$.

The possible answers are $\{(+, \text{yes}), (+, \text{no}), (-, \text{yes}), (-, \text{no})\}$, and there is only one correct answer.

(a) Histogram of pulling proportion $N_{t,1}/t$ for TaS.

(b) Proportion of runs with $N_{t,1}/t$ in the interval $(1.01 \times a, (1-a)/1.01)$ for TaS.

Figure 4: Pulling proportions of Track-and-Stop with $C$-tracking on the *Any Half-Space* problem with $K = 2$ and $a = 1/5$. Both figures use data from the same 10000 runs of Track-and-Stop. *Left*: histogram at $T = 10000$ of $N_{t,1}/t$, normalized such that the total area of the bars is 1. The values of the leftmost and rightmost bars are $\approx 15$. The probability distribution function of the Arcsine distribution is shown for comparison. *Right*: evolution over time of the mass in the interval $(1.01 \times a, \frac{1-a}{1.01})$ (non-extremal bars on the left).

The divergence from $\boldsymbol{\mu}$ to an alternative where the sign is flipped is $D_1(\boldsymbol{\mu}) = \frac{1}{2}\mu_1^2$. If $\boldsymbol{\mu}^\mathsf{T} \boldsymbol{u}^{(m)} < 0$ for both $\boldsymbol{u}^{(m)}$, then the divergence from $\boldsymbol{\mu}$ to an alternative where the second answer is yes is $D_{2,3} = \max(\frac{1}{2}(\boldsymbol{\mu}^\mathsf{T}\boldsymbol{u}^{(1)})^2, \frac{1}{2}(\boldsymbol{\mu}^\mathsf{T}\boldsymbol{u}^{(2)})^2)$. Let $D_2(\boldsymbol{\mu})$ and $D_3(\boldsymbol{\mu})$ be the two terms in that maximum. In that case, the oracle weights $\boldsymbol{w}^*(\boldsymbol{\mu})$ are

- $\{\boldsymbol{w}^{(1)}(\boldsymbol{\mu})\}$ if $D_2 > D_3$, where $\boldsymbol{w}^{(1)}(\boldsymbol{\mu}) = (\frac{D_2}{D_1+D_2}, a\frac{D_1}{D_1+D_2}, (1-a)\frac{D_1}{D_1+D_2})$,

- $\{\boldsymbol{w}^{(2)}(\boldsymbol{\mu})\}$ if $D_2 < D_3$, where $\boldsymbol{w}^{(2)}(\boldsymbol{\mu}) = (\frac{D_3}{D_1+D_3}, (1-a)\frac{D_1}{D_1+D_3}, a\frac{D_1}{D_1+D_3})$,

- the convex hull of $\{\boldsymbol{w}^{(1)}(\boldsymbol{\mu}), \boldsymbol{w}^{(2)}(\boldsymbol{\mu})\}$ if $D_2 = D_3$.

Let $\boldsymbol{\mu} = (-\mu_0, -\mu_0, -\mu_0)$ for $\mu_0 > 0$. Then $\boldsymbol{w}^*(\boldsymbol{\mu}) = \text{conv}\{(\frac{1}{2}, \frac{1}{2}a, \frac{1}{2}(1-a)), (\frac{1}{2}, \frac{1}{2}(1-a), \frac{1}{2}a)\}$.

If $a \in \{0, 1\}$, then these oracle weights correspond to $\boldsymbol{w}^{(1)}$ and $\boldsymbol{w}^{(2)}$ defined below Remark 8.

Track-and-Stop with D-tracking will track $\boldsymbol{w}^{(1)}(\hat{\boldsymbol{\mu}}_t)$ or $\boldsymbol{w}^{(2)}(\hat{\boldsymbol{\mu}}_t)$ depending on which side of the hyperplane $D_2 = D_3$ the empirical mean $\hat{\boldsymbol{\mu}}_t$ lies. As in Appendix D, we do not know the distribution of the tracked vector $\boldsymbol{w}_t$, but if $\hat{\boldsymbol{\mu}}_t$ crosses that boundary often enough, then as explained below Remark 8, D-tracking will get $N_t/t$ outside of the convex hull of $\{\boldsymbol{w}^{(1)}(\boldsymbol{\mu}), \boldsymbol{w}^{(2)}(\boldsymbol{\mu})\}$ and be suboptimal.

We verify experimentally that suboptimality in Figure 5. For these experiments, the Gaussians have variance $1/4$, $\mu_0 = 1/5$, $a = 1/10$. The fixed optimal sampling strategy samples $\text{argmin}_k N_{t,k} - t(\frac{1}{2}, \frac{1}{2}a, \frac{1}{2}(1-a))_k$.

## F    Continuity Proofs

We first introduce the necessary notions used in the modification of Berge's theorem we will apply, following [Feinberg et al., 2014].

**Definition 20.** For a function $f : U \to \mathbb{R}$ with $U$ a non-empty subset of a topological space, define the level sets

$$L_f(y, U) = \{x \in U \ : \ f(x) \leq y\},$$
$$L_f^<(y, U) = \{x \in U \ : \ f(x) < y\}.$$

Figure 5: Histogram of the stopping time of Track-and-Stop with D-tracking and of fixed optimal sampling. The data is comprised of 10000 runs of each algorithm, with $\delta = e^{-60}$.

A function $f$ is *lower semi-continuous* on $U$ if all the level sets $L_f(y, U)$ are closed. It is *inf-compact* on $U$ if all these level sets are compact. It is *upper semi-continuous* if all the strict level sets $L_f^<(y, U)$ are open.

Let $\mathbb{X}$ and $\mathbb{Y}$ be Hausdorff topological spaces. Let $u : \mathbb{X} \times \mathbb{Y} \to \mathbb{R}$ be a function, $\Phi : \mathbb{X} \to \mathbb{S}(\mathbb{Y})$ be a set-valued function, where $\mathbb{S}(\mathbb{Y})$ is the set of non-empty subsets of $\mathbb{Y}$. The objects of study are

$$v(x) = \inf_{y \in \Phi(x)} u(x, y),$$

$$\Phi^*(x) = \{y \in \Phi(x) \ : \ u(x, y) = v(x)\}.$$

For $U \subset \mathbb{X}$, let the graph of $\Phi$ restricted to $U$ be $Gr_U(\Phi) = \{(x, y) \in U \times \mathbb{Y} \ : \ y \in \Phi(x)\}$.

**Definition 21.** A function $u : \mathbb{X} \times \mathbb{Y} \to \overline{\mathbb{R}}$ is called $\mathbb{K}$-inf-compact on $Gr_{\mathbb{X}}(\Phi)$ if for all non-empty compact subset $C$ of $\mathbb{X}$, $u$ is inf-compact on $Gr_C(\Phi)$.

We will use two versions of Berge's theorem. The first one restricts $\Phi$ to be compact-valued. The second one removes that hypothesis on $\Phi$ at the price of hypotheses on $u$. Denote by $\mathbb{K}(\mathbb{X})$ the subset of $\mathbb{S}(\mathbb{X})$ containing non-empty compact subsets of $\mathbb{X}$.

**Theorem 22** (Berge's theorem)**.** *Let $\mathbb{X}$ and $\mathbb{Y}$ be Hausdorff topological spaces. Assume that*

- $\Phi : \mathbb{X} \to \mathbb{K}(\mathbb{X})$ *is continuous (i.e. both lower hemicontinuous and upper hemicontinous),*

- $u : \mathbb{X} \times \mathbb{Y} \to \mathbb{R}$ *is continuous.*

*Then the unction $v : \mathbb{X} \to \mathbb{R}$ is continuous and the solution multifunction $\Phi^* : \mathbb{X} \to \mathbb{S}(\mathbb{Y})$ is upper hemicontinuous and compact valued.*

**Theorem 23** (Feinberg et al. 2014)**.** *Assume that*

- $\mathbb{X}$ *is compactly generated,*

- $\Phi : \mathbb{X} \to \mathbb{S}(\mathbb{Y})$ *is lower hemicontinuous,*

- $u : \mathbb{X} \times \mathbb{Y} \to \mathbb{R}$ *is $\mathbb{K}$-inf-compact and upper semi-continuous on $Gr_{\mathbb{X}}(\Phi)$.*

*Then the function $v : \mathbb{X} \to \mathbb{R}$ is continuous and the solution multifunction $\Phi^* : \mathbb{X} \to \mathbb{S}(\mathbb{Y})$ is upper hemicontinuous and compact valued.*

Theorem 4 is cut into several successive lemma, whose proofs together prove the theorem. The first three lemmas prove the continuity of $(\boldsymbol{w}, \boldsymbol{\mu}) \mapsto D(\boldsymbol{w}, \boldsymbol{\mu}, \neg i)$, first in the case where $\neg i$ is compact, then in the general case.

**Lemma 24.** *Set $i \in \mathcal{I}$. If $\neg i$ is compact, then $(\boldsymbol{w}, \boldsymbol{\mu}) \mapsto D(\boldsymbol{w}, \boldsymbol{\mu}, \neg i)$ is jointly continuous on $\triangle_K \times \mathcal{M}$ and $\boldsymbol{\lambda}^*(\boldsymbol{w}, \boldsymbol{\mu})$ is non-empty, upper hemicontinuous and compact valued.*

*Proof.* We apply Theorem 22 to

- $\mathbb{X} = \triangle_K \times \mathcal{M}$,
- $\mathbb{Y} = \neg i$,
- $\Phi(\boldsymbol{\mu}) = \neg i$,
- $u((\boldsymbol{w}, \boldsymbol{\mu}), \boldsymbol{\lambda}) = D(\boldsymbol{w}, \boldsymbol{\mu}, \boldsymbol{\lambda})$.

$\Phi$ is compact-valued, non-empty and continuous (since it is constant). $u$ is continuous. The hypotheses are verified and the theorem gives the wanted result. $\qquad\square$

Let $\mathcal{Q}_+ = \{\boldsymbol{w} \in \mathbb{R}^K \; : \; \forall k \in [K], w_k \geq 0\}$, $\mathring{\mathcal{Q}}_+$ be its interior and for $\varepsilon > 0$, $\mathcal{Q}_+^\varepsilon = \{\boldsymbol{w} \in \mathbb{R}^K \; : \; \forall k \in [K], w_k \geq \varepsilon\}$.

**Lemma 25.** *The function $(\boldsymbol{w}, \boldsymbol{\mu}) \mapsto D(\boldsymbol{w}, \boldsymbol{\mu}, \neg i)$ is jointly continuous on $\mathring{\mathcal{Q}}_+ \times \mathcal{M}$. On the same set, $\boldsymbol{\lambda}^*(\boldsymbol{w}, \boldsymbol{\mu})$ is upper hemicontinuous, non-empty and compact valued. In particular, the same properties hold on $\mathring{\triangle}_K \times \mathcal{M}$.*

*Proof.* Let $\varepsilon > 0$. We prove the result for $(\boldsymbol{w}, \boldsymbol{\mu}) \in \mathcal{Q}_+^\varepsilon \times \mathcal{O}^K$ (and note that $\mathcal{M} \subseteq \mathcal{O}^K$).

We will apply Theorem 23 to

- $\mathbb{X} = \mathcal{Q}_+^\varepsilon \times \mathcal{O}^K$,
- $\mathbb{Y} = \mathcal{O}^K$,
- $\Phi((\boldsymbol{w}, \boldsymbol{\mu})) = \neg i$,
- $u((\boldsymbol{w}, \boldsymbol{\mu}), \boldsymbol{\lambda}) = D(\boldsymbol{w}, \boldsymbol{\mu}, \boldsymbol{\lambda})$,
- $v(\boldsymbol{w}, \boldsymbol{\mu}) = D(\boldsymbol{w}, \boldsymbol{\mu}, \neg i)$.

We now verify the hypothesis of the theorem. First, $\mathbb{X}$ is compactly generated since it is a metric space. Secondly, $\Phi$ is lower hemicontinuous since it is constant.

The function $u$ is continuous, hence upper semi-continuous. It remains to check that $u$ is $\mathbb{K}$-inf-compact on $Gr_{\mathbb{X}}(\Phi)$.

Let $C$ be a non-empty compact subset of $\mathcal{Q}_+^\varepsilon \times \mathcal{O}^K$. We need to prove that $u$ is inf-compact on $Gr_C(\Phi) = C \times \neg i$. The level sets $L_u(y, C \times \neg i)$ for $y \in \mathbb{R}$ are closed by continuity of $u$. Indeed they are the reverse image of a closed set $[-\infty, y]$ by a continuous function, hence they are closed in $(\mathcal{Q}_+^\varepsilon \times \mathcal{O}^K) \times \mathcal{O}^K$. We only need to prove that they are bounded.

Set $y \in \mathbb{R}$. Let $\mu_k^+ = \sup_{(\boldsymbol{w}, \boldsymbol{\mu}) \in C} \mu_k$, finite since $C$ is compact. Define $\mu_k^-$ in a similar way with an infimum. For $j \in [K]$, $\lim d(\mu_j, \lambda_j) = +\infty$ when $\lambda_j$ approaches the boundaries of the open interval $\mathcal{O}$. Then for all $k \in [K]$, there exists $\lambda_k^+$ such that $\lambda > \lambda_k^+ \Rightarrow \forall (\boldsymbol{w}, \boldsymbol{\mu}) \in C, d(\mu_k, \lambda) > y/\varepsilon$. Define $\lambda_k^-$ in a similar way. For $\boldsymbol{\lambda} \notin [\lambda_1^-, \lambda_1^+] \times \ldots \times [\lambda_K^-, \lambda_K^+]$ there exists a $k \in [K]$ such that $d(\mu_k, \lambda_k) > y/\varepsilon$ for all $(\boldsymbol{w}, \boldsymbol{\mu}) \in C$. Since $w_k \geq \varepsilon$ for all $k$, for all $(\boldsymbol{w}, \boldsymbol{\mu}) \in C$, $D(\boldsymbol{w}, \boldsymbol{\mu}, \boldsymbol{\lambda}) = \sum_{k=1}^K w_k d(\mu_k, \lambda_k) > y$. The level set $L_u(y, C \times \neg i)$ is bounded.

We have verified the hypotheses of the theorem for compacts subsets of $\mathcal{Q}_+^\varepsilon \times \mathcal{O}^K$ and obtain that $v(\boldsymbol{w}, \boldsymbol{\mu}) = D(\boldsymbol{w}, \boldsymbol{\mu}, \neg i)$ is continuous as a function of $(\boldsymbol{w}, \boldsymbol{\mu})$ on that set. On that same set, the function giving the points realizing the infimum $\boldsymbol{\lambda}^*(\boldsymbol{w}, \boldsymbol{\mu})$ is upper hemicontinuous, non-empty and compact valued. $\qquad\square$

For a projection $P$ on a subset of coordinates $S \subseteq [K]$ and $\boldsymbol{w} \in \mathcal{Q}_+$, denote the projected vector by $P\boldsymbol{w}$. The same proof as the one of the previous lemma applied to the projected spaces gives the following corollary.

**Corollary 26.** *Let $P$ be a projection on a subset of coordinates $S \subseteq [K]$. Then the function $(\boldsymbol{u}, \boldsymbol{\mu}) \mapsto D(\boldsymbol{u}, \boldsymbol{\mu}, P\neg i)$ is continuous on $P\mathring{\mathcal{Q}}_+ \times \mathcal{M}$.*

**We now extend the continuity of $D(\boldsymbol{w}, \boldsymbol{\mu}, \neg i)$ on all of $\triangle_K \times \mathcal{M}$.**

**Lemma 27.** *The function $(\boldsymbol{w}, \boldsymbol{\mu}) \mapsto D(\boldsymbol{w}, \boldsymbol{\mu}, \neg i)$ is continuous on $\triangle_K \times \mathcal{M}$.*

*Proof.* Let $(\boldsymbol{w}, \boldsymbol{\mu}) \in \triangle_K \times \mathcal{M}$ and $\boldsymbol{\lambda}^\varepsilon$ be such that $D(\boldsymbol{w}, \boldsymbol{\mu}, \boldsymbol{\lambda}) \leq D(\boldsymbol{w}, \boldsymbol{\mu}, \neg i) + \varepsilon$, which exists by definition of $D(\boldsymbol{w}, \boldsymbol{\mu}, \neg i)$ as an infimum.

The function $(\boldsymbol{w}, \boldsymbol{\mu}) \mapsto D(\boldsymbol{w}, \boldsymbol{\mu}, \boldsymbol{\lambda}^\varepsilon)$ is continuous. Hence, there exists $\varepsilon'$ such that for $\|\boldsymbol{w}' - \boldsymbol{w}\|_\infty \leq \varepsilon'$ and $\|\boldsymbol{\mu}' - \boldsymbol{\mu}\|_\infty \leq \varepsilon'$, $D(\boldsymbol{w}', \boldsymbol{\mu}', \boldsymbol{\lambda}^\varepsilon) \leq D(\boldsymbol{w}, \boldsymbol{\mu}, \boldsymbol{\lambda}^\varepsilon) + \varepsilon$. For $(\boldsymbol{w}', \boldsymbol{\mu}')$ in such a neighbourhood of $(\boldsymbol{w}, \boldsymbol{\mu})$,

$$D(\boldsymbol{w}', \boldsymbol{\mu}', \neg i) \leq D(\boldsymbol{w}', \boldsymbol{\mu}', \boldsymbol{\lambda}^\varepsilon) \leq D(\boldsymbol{w}, \boldsymbol{\mu}, \boldsymbol{\lambda}^\varepsilon) + \varepsilon \leq D(\boldsymbol{w}, \boldsymbol{\mu}, \neg i) + 2\varepsilon \ .$$

We proved that $(\boldsymbol{w}, \boldsymbol{\mu}) \mapsto D(\boldsymbol{w}, \boldsymbol{\mu}, \neg i)$ is upper semi-continuous.

Now let $\varepsilon > 0$ be such that $\min_{k:w_k>0} w_k \geq 2\varepsilon$ and $(\boldsymbol{w}', \boldsymbol{\mu}')$ be in an $\varepsilon$ neighbourhood of $(\boldsymbol{w}, \boldsymbol{\mu})$. Then $w_k > 0 \Rightarrow w_k' > \varepsilon$. For $\mathbf{u} \in \triangle_K$, denote by $P\mathbf{u}$ its projection on the coordinates for which $w_k > 0$.

$$D(\boldsymbol{w}', \boldsymbol{\mu}', \neg i) \geq D(P\boldsymbol{w}', \boldsymbol{\mu}', P\neg i) \ ,$$
$$D(\boldsymbol{w}, \boldsymbol{\mu}, \neg i) = D(P\boldsymbol{w}, \boldsymbol{\mu}, P\neg i) \ .$$

By the previous lemma, $(P\boldsymbol{w}', \boldsymbol{\mu}') \mapsto D(P\boldsymbol{w}', \boldsymbol{\mu}', P\neg i)$ is continuous on $P\mathcal{Q}_+^\varepsilon \times \mathcal{M}$. Hence for $(\boldsymbol{w}', \boldsymbol{\mu}')$ in a small enough neighbourhood of $(\boldsymbol{w}, \boldsymbol{\mu})$, $D(P\boldsymbol{w}', \boldsymbol{\mu}', P\neg i) \geq D(P\boldsymbol{w}, \boldsymbol{\mu}, P\neg i) - \varepsilon$. In that neighbourhood,

$$D(\boldsymbol{w}', \boldsymbol{\mu}', \neg i) \geq D(P\boldsymbol{w}', \boldsymbol{\mu}', P\neg i) \geq D(P\boldsymbol{w}, \boldsymbol{\mu}, P\neg i) - \varepsilon \geq D(\boldsymbol{w}, \boldsymbol{\mu}, \neg i) - \varepsilon \ .$$

This proves the lower semi-continuity of $(\boldsymbol{w}, \boldsymbol{\mu}) \mapsto D(\boldsymbol{w}, \boldsymbol{\mu}, \neg i)$. We now have both lower and upper semi-continuity: that function is continuous.

$\square$

We proved continuity of the function of $\triangle_K \times \mathcal{M}$ but upper hemi-continuity of $\boldsymbol{\lambda}^*$ only for $\boldsymbol{w} \in \mathring{\triangle}_K$. We now show an example where $\boldsymbol{\lambda}^*$ is empty at a $\boldsymbol{w}$ on the boundary of the simplex.

Let $\boldsymbol{\mu} = (0, 0)$, $\neg i = \{(\sqrt{x}, \sqrt{1 + 1/x}) \ : \ x \in \mathbb{R}^+\}$ and $d(\mu_i, \lambda_i) = (\mu_i - \lambda_i)^2$. Then

$$\inf_{\boldsymbol{\lambda}} [w_1 d(\mu_1, \lambda_1) + w_2 d(\mu_2, \lambda_2)] = \inf_x (w_1 x + (1 - w_1)(1 + \frac{1}{x})) = 1 - w_1 + 2\sqrt{w_1(1 - w_1)} \ ,$$

with minimum attained for $w_1 \in (0, 1)$ at $x = \sqrt{\frac{1 - w_1}{w_1}}$. Hence for $w_1 > 0$, $\boldsymbol{\lambda}^*(\boldsymbol{w}, (0, 0)) = \{(\sqrt{\frac{1 - w_1}{w_1}}, \sqrt{\frac{w_1}{1 - w_1}})\}$, non-empty and compact. If $w_1 = 0$ then the infimum is not attained and $\boldsymbol{\lambda}^*$ is empty.

**Lemma 28.** *For all $i \in \mathcal{I}$, $D(\boldsymbol{\mu}, \neg i)$ is a continuous function of $\boldsymbol{\mu}$ on $\mathcal{M}$ and $\boldsymbol{w}^*(\boldsymbol{\mu}, \neg i)$ is upper hemicontinuous and compact-valued.*

*Proof.* We apply Theorem 22 to

- $\mathbb{X} = \mathcal{M}$,
- $\mathbb{Y} = \triangle_K$,
- $\Phi(\boldsymbol{\mu}) = \triangle_K$,
- $u(\boldsymbol{\mu}, \boldsymbol{w}) = D(\boldsymbol{w}, \boldsymbol{\mu}, \neg i)$.

$\Phi$ is compact-valued, non-empty and continuous (since it is constant). $u$ is continuous by Lemma 27. The hypotheses are verified and the theorem gives the wanted result. $\square$

**Lemma 29.** *For all $i \in \mathcal{I}$, $D(\boldsymbol{\mu})$ is a continuous function of $\boldsymbol{\mu}$ on $\mathcal{M}$ and $\boldsymbol{w}^*(\boldsymbol{\mu})$ is upper hemicontinuous and compact-valued.*

*Proof.* We apply Theorem 22 to

- $\mathbb{X} = \mathcal{M}$,
- $\mathbb{Y} = \triangle_K$,
- $\Phi(\boldsymbol{\mu}) = \triangle_K$,
- $u(\boldsymbol{\mu}, \boldsymbol{w}) = \max_{i \in \mathcal{I}} D(\boldsymbol{w}, \boldsymbol{\mu}, \neg i)$.

$\Phi$ is compact-valued, non-empty and continuous (since it is constant). $u$ is continuous since it is a finite maximum of continuous functions. The hypotheses are verified and the theorem gives the wanted result. $\square$

**Lemma 30.** *$i_F(\boldsymbol{\mu})$ is upper hemicontinuous and compact valued.*

*Proof.* We apply Theorem 22 to

- $\mathbb{X} = \mathcal{M}$,
- $\Phi(\boldsymbol{\mu}) = \{1, \ldots, K\}$,
- $\mathbb{Y} = \{1, \ldots, K\}$,
- $u(\boldsymbol{\mu}, \boldsymbol{w}) = D(\boldsymbol{\mu}, \neg i)$.

$\Phi$ is compact-valued, non-empty and continuous (since it is constant). $u$ is continuous Lemma 28. The hypotheses are verified and the theorem gives the wanted result. $\qquad\square$

# G   Algorithm Analysis

## G.1   Proof of Lemma 14

**Lemma 31** (Lemma 19 of Garivier and Kaufmann 2016). *There exists two constants $B$ and $C$ (that depend on $\mu$ and $\xi$) such that*

$$\mathbb{P}_\mu(\mathcal{E}_T'^{\,c}) \leq BTe^{-C\sqrt{h(T)}} \ .$$

For $h(T) = \sqrt{T}$, $\sum_{T=1}^{+\infty} \mathbb{P}_\mu(\mathcal{E}_T'^{\,c})$ is then finite.

**Lemma 32** (Magureanu et al. 2014). *Let $\beta(t,\delta) = \log(Ct^2/\delta)$ with $C$ a constant verifying the inequality $C \geq e\sum_{t=1}^{\infty}(\frac{e}{K})^K\frac{(\log^2(Ct^2)\log(t))^K}{t^2}$ . Then*

$$\mathbb{P}_{\boldsymbol{\mu}}\left\{\exists t \in \mathbb{N}, \ \sum_{k=1}^K N_{t,k}d(\hat{\mu}_{t,k}, \mu_k) > \beta(t,\delta)\right\} \leq \delta \ .$$

Let $f(t) = \exp(\beta(t, 1/t^5)) = Ct^{10}$ in the definition of the confidence ellipsoid $\mathcal{C}_t$ (see Algorithm 1). Then

$$\sum_{T=T_0}^{+\infty} \mathbb{P}_{\boldsymbol{\mu}}(\mathcal{E}_T^c) \leq \sum_{T=1}^{+\infty} \sum_{t=\sqrt{T}}^{T} \frac{1}{t^5} < +\infty \ .$$

## G.2   Proof of Lemma 15

Set $T > T_1$ and suppose that $\mathcal{E}_T \cap \mathcal{E}_T'$ is true. For $t \geq \eta(T)$, if $\tau_\delta > t$ then $tC_{\varepsilon,\xi}^*(\boldsymbol{\mu}) \leq D(N_t, \hat{\boldsymbol{\mu}}_t, \neg i_{\boldsymbol{\mu}}) \leq \beta(t,\delta)$ , hence $t \leq \beta(t,\delta)/C_{\varepsilon;\xi}^*(\boldsymbol{\mu})$ .

$$
\begin{aligned}
\min(\tau_\delta, T) &\leq \lceil\eta(T)\rceil + \sum_{t=\lceil\eta(T)\rceil+1}^{T} \mathbb{I}\{\tau_\delta > t-1\} \\
&\leq \lceil\eta(T)\rceil + \sum_{t=\lceil\eta(T)\rceil+1}^{T} \mathbb{I}\{t \leq 1 + \frac{\beta(t,\delta)}{C_{\varepsilon,\xi}^*(\boldsymbol{\mu})}\} \\
&\leq \lceil\eta(T)\rceil + \sum_{t=\lceil\eta(T)\rceil+1}^{T} \mathbb{I}\{t \leq 1 + \frac{\beta(T,\delta)}{C_{\varepsilon,\xi}^*(\boldsymbol{\mu})}\} \\
&\leq \max\left(\lceil\eta(T)\rceil, 1 + \frac{\beta(T,\delta)}{C_{\varepsilon,\xi}^*(\boldsymbol{\mu})}\right) \ .
\end{aligned}
$$

Suppose that $\tau_\delta > T$. Then $T \leq \max\left(\lceil\eta(T)\rceil, 1 + \frac{\beta(T,\delta)}{C_{\varepsilon,\xi}^*(\boldsymbol{\mu})}\right)$ . But by hypothesis, $\eta(T) < T-1$, such that that inequality implies $T \leq 1 + \frac{\beta(T,\delta)}{C_{\varepsilon,\xi}^*(\boldsymbol{\mu})}$ . We conclude that

$$\tau_\delta \leq \inf\left\{T : T > 1 + \frac{\beta(T,\delta)}{C_{\varepsilon,\xi}^*(\boldsymbol{\mu})}\right\} \ .$$

### G.3 Continuity Results and Proof of Lemma 6

**Lemma 33.** *Let $\varepsilon > 0$ and $A \subseteq \triangle_K$ be a convex set and let $\boldsymbol{w}_1, \ldots, \boldsymbol{w}_t \in \triangle_K$ be such that for all $s \in [t]$, $\inf_{\boldsymbol{w} \in A} \|\boldsymbol{w}_s - \boldsymbol{w}\|_\infty \leq \varepsilon$. Then $\inf_{\boldsymbol{w} \in A} \|\frac{1}{t} \sum_{s=1}^t \boldsymbol{w}_s - \boldsymbol{w}\|_\infty \leq \varepsilon$.*

*Proof.* For $s \in [t]$, let $\boldsymbol{w}_s^* \in A$ be such that $\|\boldsymbol{w}_s - \boldsymbol{w}_s^*\|_\infty \leq \varepsilon$. Then $\frac{1}{t} \sum_{s=1}^t \boldsymbol{w}_s^* \in A$ by convexity and

$$\|\frac{1}{t}\sum_{s=1}^t \boldsymbol{w}_s - \frac{1}{t}\sum_{s=1}^t \boldsymbol{w}_s^*\|_\infty \leq \frac{1}{t}\sum_{s=1}^t \|\boldsymbol{w}_s - \boldsymbol{w}_s^*\|_\infty \leq \varepsilon.$$

$\square$

*Proof of Lemma 6.* Let $\varepsilon > 0$. By Theorem 4, $\boldsymbol{w}^*$ is upper hemicontinuous: there exists $\xi > 0$ such that if $\|\hat{\boldsymbol{\mu}}_t - \boldsymbol{\mu}\|_\infty \leq \xi$ then for all $\boldsymbol{w}_t \in \boldsymbol{w}^*(\hat{\boldsymbol{\mu}}_t)$, $\inf_{\boldsymbol{w} \in \boldsymbol{w}^*(\boldsymbol{\mu})} \|\boldsymbol{w}_t - \boldsymbol{w}\|_\infty \leq \varepsilon$.

For $\xi > 0$, there exists $t_\xi$ such that for $t \geq t_\xi$, $\|\hat{\boldsymbol{\mu}}_t - \boldsymbol{\mu}\|_\infty \leq \xi$ by hypothesis. Then for $\boldsymbol{w} \in \boldsymbol{w}^*(\boldsymbol{\mu})$,

$$\left\|\frac{1}{t}\sum_{s=1}^t \boldsymbol{w}_s - \boldsymbol{w}\right\|_\infty \leq \frac{t_\xi}{t} + \frac{t - t_\xi}{t}\left\|\frac{1}{t - t_\xi}\sum_{s=t_\xi}^t \boldsymbol{w}_s - \boldsymbol{w}\right\|_\infty.$$

Taking infimums and using the convexity of $\boldsymbol{w}^*(\boldsymbol{\mu})$ to apply Lemma 33,

$$\inf_{\boldsymbol{w} \in \boldsymbol{w}^*(\boldsymbol{\mu})}\left\|\frac{1}{t}\sum_{s=1}^t \boldsymbol{w}_s - \boldsymbol{w}\right\|_\infty \leq \frac{t_\xi}{t} + \varepsilon \leq 2\varepsilon \text{ for } t \geq t_\xi/\varepsilon.$$

$\square$

### G.4 Proof of Lemma 16

Under $\mathcal{E}_T$, for $t \geq h(T)$, $\boldsymbol{\mu} \in \mathcal{C}_t$. Hence $i_F(\boldsymbol{\mu}) \subseteq I_t$.

For $\boldsymbol{\mu}, \boldsymbol{\mu}' \in \mathcal{M}$, let $ch(\boldsymbol{\mu}, \boldsymbol{\mu}') = \inf_{\boldsymbol{\lambda} \in \mathbb{R}^K} \sum_{k=1}^K (d(\lambda_k, \mu_k) + d(\lambda_k, \mu_k'))$. $ch$ is a semi-metric on $\mathcal{M}$.

Suppose that the event $\mathcal{E}_T$ holds and set $t > h(T)$. Then on one hand $\boldsymbol{\mu} \in \mathcal{C}_t$, such that $D(N_{t-1}, \hat{\boldsymbol{\mu}}_{t-1}, \boldsymbol{\mu}) \leq \log f(t - 1)$. On the other hand, by definition every point $\boldsymbol{\mu}' \in \mathcal{C}_t$ verifies $D(N_{t-1}, \hat{\boldsymbol{\mu}}_{t-1}, \boldsymbol{\mu}') \leq \log f(t - 1)$. We obtain in particular that

$$\sum_{k=1}^K N_{t-1,k}(d(\hat{\mu}_{t-1,k}, \mu_k) + d(\hat{\mu}_{t-1,k}, \mu_k')) \leq 2\log f(t - 1).$$

By hypotheses, $N_{t-1,k} \geq n(t-1)$. We obtain that $ch(\boldsymbol{\mu}, \boldsymbol{\mu}') \leq \frac{2\log f(t-1)}{n(t-1)}$ for all $\boldsymbol{\mu}' \in \mathcal{C}_t$.

By upper hemicontinuity of $i_F(\boldsymbol{\mu})$, there exists $\varepsilon > 0$ such that $\|\boldsymbol{\mu} - \boldsymbol{\mu}'\|_\infty \leq \varepsilon \Rightarrow i_F(\boldsymbol{\mu}') \subseteq i_F(\boldsymbol{\mu})$. There exists $\Delta > 0$ such that $ch(\boldsymbol{\mu}, \boldsymbol{\mu}') \leq \Delta \Rightarrow i_F(\boldsymbol{\mu}') \subseteq i_F(\boldsymbol{\mu})$.

For such a $\Delta$ and $T_\Delta = \inf\{t \in \mathbb{N} : \frac{2\log f(t)}{n(t)} \leq \Delta\}$, if $t \geq \max(h(T), T_\Delta)$, then $i_F(\boldsymbol{\mu}') \subseteq i_F(\boldsymbol{\mu})$ for all $\boldsymbol{\mu}' \in \mathcal{C}_t$. hence $I_t = \bigcup_{\boldsymbol{\mu}' \in \mathcal{C}_t} i_F(\boldsymbol{\mu}') \subseteq i_F(\boldsymbol{\mu})$.

### G.5 Proof of Lemma 17

**Lemma 34** (Garivier and Kaufmann 2016). *For all $t \geq 1$ and $k \in [K]$, the C-tracking rule ensures that $N_{t,k} \geq \sqrt{t + K^2} - 2K$ and that*

$$\left\|N_t - \sum_{s=0}^{t-1} \boldsymbol{w}_s\right\|_\infty \leq K(1 + \sqrt{t}).$$

**Lemma 35.** *Suppose that there exists $T_I \in \mathbb{N}$ such that for $T \geq T_I$, $\boldsymbol{w}_t \in \boldsymbol{w}^*(\hat{\boldsymbol{\mu}}_t, \neg i_{\boldsymbol{\mu}})$. Then for $T$ such that $h(T) \geq T_I$, it holds that on $\mathcal{E}'_T$ C-Tracking verifies*

$$\forall t \geq 4\frac{K^2}{\varepsilon^2} + 3\frac{h(T)}{\varepsilon}, \quad \inf_{\boldsymbol{w} \in \boldsymbol{w}^*(\boldsymbol{\mu}, \neg i_{\boldsymbol{\mu}})} \left\| \frac{N(t)}{t} - \boldsymbol{w} \right\|_\infty \leq 3\varepsilon .$$

*Proof.* Suppose that $T$ verifies $h(T) \geq T_I$. Using Lemma 34, for $t \geq h(T)$ one can write for all $\boldsymbol{w} \in \boldsymbol{w}^*(\boldsymbol{\mu}, \neg i_{\boldsymbol{\mu}})$,

$$\left\| \frac{N_t}{t} - \boldsymbol{w} \right\|_\infty \leq \left\| \frac{N_t}{t} - \frac{1}{t}\sum_{s=0}^{t-1} \boldsymbol{w}_s \right\|_\infty + \left\| \frac{1}{t}\sum_{s=0}^{t-1} \boldsymbol{w}_s - \boldsymbol{w} \right\|_\infty$$

$$\leq \frac{K(1+\sqrt{t})}{t} + \left\| \frac{1}{t}\sum_{s=0}^{t-1} \boldsymbol{w}_s - \boldsymbol{w} \right\|_\infty$$

$$\leq \frac{2K}{\sqrt{t}} + \frac{h(T)}{t} + \left\| \frac{1}{t}\sum_{s=h(T)}^{t-1} (\boldsymbol{w}_s - \boldsymbol{w}) \right\|_\infty .$$

The definition of event $\mathcal{E}'_T$ uses $\xi > 0$ such that if $\|\hat{\boldsymbol{\mu}}_t - \boldsymbol{\mu}\|_\infty \leq \xi$ then for all $\boldsymbol{w}_t \in \boldsymbol{w}^*(\hat{\boldsymbol{\mu}}_t, \neg i_{\boldsymbol{\mu}})$, $\inf_{\boldsymbol{w} \in \boldsymbol{w}^*(\boldsymbol{\mu}, \neg i_{\boldsymbol{\mu}})} \|\boldsymbol{w}_t - \boldsymbol{w}\|_\infty \leq \varepsilon$. Under that event, for $t \geq h(T)$,

$$\|\hat{\boldsymbol{\mu}}_t - \boldsymbol{\mu}\|_\infty \leq \xi ,$$
$$\forall \boldsymbol{w}_t \in \boldsymbol{w}^*(\hat{\boldsymbol{\mu}}_t, \neg i_{\boldsymbol{\mu}}), \quad \inf_{\boldsymbol{w} \in \boldsymbol{w}^*(\boldsymbol{\mu}, \neg i_{\boldsymbol{\mu}})} \|\boldsymbol{w}_t - \boldsymbol{w}\|_\infty \leq \varepsilon .$$

The convexity of $\boldsymbol{w}^*(\boldsymbol{\mu}, \neg i_{\boldsymbol{\mu}})$ ensures that $\inf_{\boldsymbol{w} \in \boldsymbol{w}^*(\boldsymbol{\mu}, \neg i_{\boldsymbol{\mu}})} \|\frac{1}{t}\sum_{s=h(T)}^T \boldsymbol{w}_s - \boldsymbol{w}\|_\infty \leq \varepsilon$ as well by Lemma 33. Hence, taking infimums and using the hypothesis that the event $\mathcal{E}'_T$ holds,

$$\inf_{\boldsymbol{w} \in \boldsymbol{w}^*(\boldsymbol{\mu}, \neg i_{\boldsymbol{\mu}})} \left\| \frac{N_t}{t} - \boldsymbol{w} \right\|_\infty \leq \frac{2K}{\sqrt{t}} + \frac{h(T)}{t} + \inf_{\boldsymbol{w} \in \boldsymbol{w}^*(\boldsymbol{\mu}, \neg i_{\boldsymbol{\mu}})} \left\| \frac{1}{t}\sum_{s=h(T)}^{t-1} (\boldsymbol{w}_s - \boldsymbol{w}) \right\|_\infty$$

$$\leq \frac{2K}{\sqrt{t}} + \frac{h(T)}{t} + \varepsilon .$$

For $t \geq 2\frac{K^2}{\varepsilon^2}\left(1 + \varepsilon\frac{h(T)}{2K^2} + \sqrt{1 + \varepsilon\frac{h(T)}{K^2}}\right)$, the right-hand-side is smaller than $2\varepsilon$. In particular, this is also true for $t \geq 4\frac{K^2}{\varepsilon^2} + 3\frac{h(T)}{\varepsilon}$. $\qquad\square$

*Proof of Lemma 17.* Let $T \in \mathbb{N}$ be such that $h(T) \geq T_\Delta$. Under $\mathcal{E}_T$, for $t \geq h(T)$ the set $I_t$ is constant and equal to $i_F(\boldsymbol{\mu})$ by Lemma 16. For this stage on, $i_t = i_{\boldsymbol{\mu}}$ is constant and $\boldsymbol{w}^*(\boldsymbol{\mu}, \neg i_t) \subseteq \boldsymbol{w}^*(\boldsymbol{\mu})$. We can hence take $T_I = h(T)$ in Lemma 35 and we get the wanted result. $\qquad\square$

### G.6 Proof of the empirical complexity of Track and Stop

We prove Theorem 7 and Theorem 9.

Lemma 16 depends only on the amount of forced exploration, thus it is valid for Track and Stop. After some $T_\Delta > 0$, $I_t = i_F(\boldsymbol{\mu})$. Since $\hat{\boldsymbol{\mu}}_t \in I_t$, $i_F(\hat{\boldsymbol{\mu}}_t) \subseteq i_F(\boldsymbol{\mu})$. The alternative selected by Track and Stop to compute $\boldsymbol{w}_t$ will be in $i_F(\boldsymbol{\mu})$.

We modify the event $\mathcal{E}'_T$ used for Sticky Track and Stop into $\mathcal{E}'_T = \bigcap_{t=h(T)}^T \{\|\hat{\boldsymbol{\mu}}_t - \boldsymbol{\mu}\|_\infty \leq \xi\}$ where $\xi$ is such that

$$\|\boldsymbol{\mu}' - \boldsymbol{\mu}\|_\infty \leq \xi \Rightarrow \forall \boldsymbol{w}' \in \boldsymbol{w}^*(\boldsymbol{\mu}') \, \exists \boldsymbol{w} \in \boldsymbol{w}^*(\boldsymbol{\mu}), \, \|\boldsymbol{w}' - \boldsymbol{w}\|_\infty \leq \varepsilon .$$

The difference is that we use the upper hemicontinuity of $\boldsymbol{w}^*(\boldsymbol{\mu})$ instead of $\boldsymbol{w}^*(\boldsymbol{\mu}, \neg i)$ for some $i \in \mathcal{I}$.

From that point on, we proceed as in the proof of the sample complexity of Sticky Track and Stop, except that $N_t/t$ will not necessarily converge to $\boldsymbol{w}^*(\boldsymbol{\mu}, i_{\boldsymbol{\mu}})$, but to $\text{conv}(\boldsymbol{w}^*(\boldsymbol{\mu}))$, convex hull of $\boldsymbol{w}^*(\boldsymbol{\mu})$. Lemma 17 is true for Track and Stop with the adapted $\mathcal{E}'_T$ if $\boldsymbol{w}^*(\boldsymbol{\mu}, i_{\boldsymbol{\mu}})$ is replaced by $\text{conv}(\boldsymbol{w}^*(\boldsymbol{\mu}))$. The analogue of $C^*_{\varepsilon,\xi}(\boldsymbol{\mu})$ is

$$C_{\varepsilon,\xi}(\boldsymbol{\mu}) = \inf_{\substack{\boldsymbol{\mu}':\|\boldsymbol{\mu}'-\boldsymbol{\mu}\|_\infty \leq \xi \\ \boldsymbol{w}':\inf_{\boldsymbol{w}\in\text{conv}(\boldsymbol{w}^*(\boldsymbol{\mu}))}\|\boldsymbol{w}'-\boldsymbol{w}\|_\infty \leq 3\varepsilon}} D(\boldsymbol{w}',\boldsymbol{\mu}') .$$

Let $T_\Delta$ be defined as in Lemma 16. Let $T$ be such that $h(T) \geq T_\Delta$. Let $\eta(T) = 4\frac{K^2}{\varepsilon^2} + 3\frac{h(T)}{\varepsilon}$ . By Lemma 17 changed as explained, for $t \geq \eta(T)$, if $\mathcal{E}_T \cap \mathcal{E}'_T$ then $D(N_t, \hat{\boldsymbol{\mu}}_t) \geq tC_{\varepsilon,\xi}(\boldsymbol{\mu})$.

We now apply Lemma 15. $\eta(T) < T - 1$ if $h(T) < \frac{\varepsilon}{3}(T-1) - \frac{4}{3}\frac{K^2}{\varepsilon}$. For $h(T) = \sqrt{T}$ and $T$ bigger than a constant $T_\eta$ depending on $K$ and $\varepsilon$, this is true. Then under $\mathcal{E}_T \cap \mathcal{E}'_T$, the hypotheses of Lemma 15 are verified with $T_1 = h^{-1}(\max(T_\Delta, T_\eta))$.

We obtain that the hypotheses of Lemma 13 are verified for

$$T_0 = \max(T_1, \inf\{T : 1 + \frac{\beta(T,\delta)}{C_{\varepsilon,\xi}(\boldsymbol{\mu})} \leq T\}) .$$

Note that $\lim_{\delta\to 0} T_0 / \log(1/\delta) = 1/C_{\varepsilon,\xi}(\boldsymbol{\mu})$. Taking $\varepsilon \to 0$ (hence $\xi \to 0$ as well), we obtain $\lim_{\delta\to 0} \frac{\mathbb{E}_{\boldsymbol{\mu}}[\tau_\delta]}{\log(1/\delta)} = \frac{1}{\lim_{\varepsilon\to 0} C_{\varepsilon,\xi}(\boldsymbol{\mu})}$.

Finally, $\lim_{\delta\to 0} C_{\varepsilon,\xi}(\boldsymbol{\mu}) = \inf_{\boldsymbol{w}\in\text{conv}(\boldsymbol{w}^*(\boldsymbol{\mu}))} D(\boldsymbol{w},\boldsymbol{\mu})$. This proves Theorem 9.

If $i_F(\boldsymbol{\mu})$ is a singleton, then $\boldsymbol{w}^*(\boldsymbol{\mu})$ is convex and $\text{conv}(\boldsymbol{w}^*(\boldsymbol{\mu})) = \boldsymbol{w}^*(\boldsymbol{\mu})$, leading to the observation that $\inf_{\boldsymbol{w}\in\text{conv}(\boldsymbol{w}^*(\boldsymbol{\mu}))} D(\boldsymbol{w},\boldsymbol{\mu}) = D(\boldsymbol{\mu})$. In that case, Track-and-Stop is asymptotically optimal: Theorem 7 is proved.

# H   Divergences

An important building block in pure exploration algorithms is the largest weighted distance from $\boldsymbol{\mu}$ to the closest point $\boldsymbol{\lambda}$ in some set of alternatives,

$$D(\boldsymbol{\mu}, \Lambda) = \sup_{\boldsymbol{w}\in\triangle} \inf_{\boldsymbol{\lambda}\in\Lambda} \sum_k w_k d(\mu_k, \lambda_k)$$

In this section we compute a few of these distances in closed form to get a feeling for their behaviour. We do it for the Gaussian divergence $d(\mu, \lambda) = \frac{1}{2}(\mu - \lambda)^2$.

## H.1   Hyper-planes and Half-spaces

**Lemma 36.** *When* $\Lambda = \left\{\boldsymbol{\lambda} \in \mathbb{R}^K \,\middle|\, \langle \boldsymbol{a}, \boldsymbol{\lambda} \rangle = b\right\}$ *is a hyper-plane, we find*

$$D(\boldsymbol{\mu}, \Lambda) = \frac{1}{2}\left(\frac{\langle \boldsymbol{a}, \boldsymbol{\mu} \rangle - b}{\sum_{i=1}^d |a_i|}\right)^2 \qquad and \qquad w_i^*(\boldsymbol{\mu}) = \frac{|a_i|}{\sum_{i=1}^d |a_i|}.$$

Note that the same result holds for the half-space $\Lambda = \left\{\boldsymbol{\lambda} \in \mathbb{R}^K \,\middle|\, \langle \boldsymbol{a}, \boldsymbol{\lambda} \rangle \geq b\right\}$ when $\boldsymbol{\mu} \notin \Lambda$, i.e. $\langle \boldsymbol{a}, \boldsymbol{\mu} \rangle < b$. If $\boldsymbol{\mu} \in \Lambda$ then $D(\boldsymbol{\mu}, \Lambda) = 0$. The Lemma implies in particular that for Best Arm Identification with $K = 2$ arms, corresponding to $\boldsymbol{a} = (-1, +1)$ and $b = 0$, the optimal weights $\boldsymbol{w}^*$ are uniform, as was shown by Kaufmann et al. [2016]. For the $\epsilon$-BAI variant of the problem we set $b = \pm\epsilon$, so here $\boldsymbol{w}^*$ is also uniform.

*Proof.* We have

$$D(\boldsymbol{\mu}, \Lambda) = \sup_{\boldsymbol{w}\in\triangle} \inf_{\boldsymbol{\lambda}:\langle\boldsymbol{a},\boldsymbol{\lambda}\rangle=b} \sum_{i=1}^d w_i \frac{(\mu_i - \lambda_i)^2}{2}$$

Introducing Lagrange multiplier $\theta$, we find

$$\sup_{\boldsymbol{w}\in\triangle} \sup_{\theta} \inf_{\boldsymbol{\lambda}\in\mathbb{R}^d} \sum_{i=1}^{d} w_i \frac{(\mu_i - \lambda_i)^2}{2} - \theta\left(\langle \boldsymbol{a}, \boldsymbol{\lambda}\rangle - b\right)$$

Plugging in the solution $\lambda_i = \mu_i + \frac{\theta a_i}{w_i}$ results in

$$\sup_{\boldsymbol{w}\in\triangle} \sup_{\theta} -\theta^2 \sum_{i=1}^{d} \frac{a_i^2}{2 w_i} - \theta\left(\langle \boldsymbol{a}, \boldsymbol{\mu}\rangle - b\right)$$

Now solving for $\theta$ results in $\theta = \frac{-(\langle \boldsymbol{a}, \boldsymbol{\mu}\rangle - b)}{\sum_{i=1}^{d} \frac{a_i^2}{w_i}}$ and objective function value

$$\sup_{\boldsymbol{w}\in\triangle} \frac{\left(\langle \boldsymbol{a}, \boldsymbol{\mu}\rangle - b\right)^2}{2 \sum_{i=1}^{d} \frac{a_i^2}{w_i}}$$

Further solving for $\boldsymbol{w}$ tells us that $w_i = \frac{|a_i|}{\sum_{i=1}^{d} |a_i|}$ and hence the value is

$$\frac{1}{2}\left(\frac{\langle \boldsymbol{a}, \boldsymbol{\mu}\rangle - b}{\sum_{i=1}^{d} |a_i|}\right)^2.$$

$\square$

## H.2 Minimum Threshold

The following two lemmas appear as [Kaufmann et al., 2018, Lemma 1] for general divergences $d(\mu, \lambda)$.

**Lemma 37.** *Let* $\Lambda = \left\{\boldsymbol{\lambda} \in \mathbb{R}^K \middle| \min_k \lambda_k \leq \gamma \right\}$. *Then when* $\boldsymbol{\mu} \notin \Lambda$,

$$D(\boldsymbol{\mu}, \Lambda) = \frac{1}{\sum_k \frac{1}{d(\mu_a, \gamma)}} \qquad \text{where} \qquad w_k^*(\boldsymbol{\mu}) = \frac{\frac{1}{d(\mu_k, \gamma)}}{\sum_j \frac{1}{d(\mu_j, \gamma)}}$$

**Lemma 38.** *Let* $\Lambda = \left\{\boldsymbol{\lambda} \in \mathbb{R}^K \middle| \min_k \lambda_k \geq \gamma \right\}$. *Then when* $\boldsymbol{\mu} \notin \Lambda$,

$$D(\boldsymbol{\mu}, \Lambda) = d\left(\min_k \mu_k, \gamma\right) \qquad \text{where} \qquad \boldsymbol{w}^*(\boldsymbol{\mu}) = \mathbf{1}_{k=\operatorname{argmin}_j \mu_j}.$$

For the version of the problem with slack $\epsilon$, we can simply replace $\gamma$ by the appropriate $\gamma \pm \epsilon$.

## H.3 Sphere

We now consider the distance to the sphere both from within and from the outside.

**Lemma 39.** *Let* $\Lambda = \left\{\boldsymbol{\lambda} \in \mathbb{R}^K \middle| \|\boldsymbol{\lambda}\| = 1 \right\}$. *Consider any* $\boldsymbol{\mu} \in \mathbb{R}^K$. *Then*

$$D(\boldsymbol{\mu}, \Lambda) = \frac{1}{2K^2}\left(\sqrt{K\left(1 - \|\boldsymbol{\mu}\|^2\right) + \|\boldsymbol{\mu}\|_1^2} - \|\boldsymbol{\mu}\|_1\right)^2$$

$$\text{and} \quad w_k^*(\boldsymbol{\mu}) = \frac{1}{K} + \frac{|\mu_k| - \frac{1}{K}\|\boldsymbol{\mu}\|_1}{\sqrt{K\left(1 - \|\boldsymbol{\mu}\|^2\right) + \|\boldsymbol{\mu}\|_1^2}},$$

*provided that*

$$\left(\|\boldsymbol{\mu}\|_1 - K \min_k |\mu_k|\right)^2 \leq \|\boldsymbol{\mu}\|_1^2 - K\left(\|\boldsymbol{\mu}\|^2 - 1\right) \tag{5}$$

Note that the proviso is always satisfied when $\|\boldsymbol{\mu}\| \leq 1$. When $\|\boldsymbol{\mu}\| > 1$ it depends. When the proviso is not satisfied, boundary conditions are active. In that case a pairwise swapping argument shows that $w_k^*(\boldsymbol{\mu}) = 0$ for the $k$ of minimal $|\mu_k|$. The rest of the solution is found by removing $k$, and solving the remaining problem of size $K - 1$.

*Proof.* We need to find

$$D(\boldsymbol{\mu}, \Lambda) = \max_{\boldsymbol{w} \in \triangle} \min_{\boldsymbol{\lambda}:\|\boldsymbol{\lambda}\|=1} \frac{1}{2} \sum_k w_k (\mu_k - \lambda_k)^2$$

As strong duality holds for the inner problem [Boyd and Vandenberghe, 2004, Appendix B], we may introduce a Lagrange multiplier $\theta$ for the constraint, and write

$$\max_{\boldsymbol{w} \in \triangle, \theta} \min_{\boldsymbol{\lambda} \in \mathbb{R}^K} \frac{1}{2} \sum_k w_k (\mu_k - \lambda_k)^2 + \frac{\theta}{2}\left(1 - \|\boldsymbol{\lambda}\|^2\right).$$

The innermost problem is unbounded in $\boldsymbol{\lambda}$ unless $\min_k w_k \geq \theta$, so we add this as an outer constraint. Then the minimiser is found at $\lambda_k = \frac{\mu_k}{1 - \frac{\theta}{w_k}}$, and by substituting that in, the problem simplifies to

$$\max_{\substack{\boldsymbol{w} \in \triangle \\ \theta \leq \min_k w_k}} -\frac{1}{2} \sum_k \frac{\mu_k^2}{\frac{1}{\theta} - \frac{1}{w_k}} + \frac{\theta}{2} \quad = \quad \max_{\substack{\boldsymbol{w} \in \triangle \\ \theta \leq \min_k w_k}} -\frac{1}{2} \sum_k \frac{\mu_k^2 \theta^2}{w_k - \theta} + \frac{\theta}{2}\left(1 - \|\boldsymbol{\mu}\|^2\right).$$

As a point of interpretation, note that we will find $\theta > 0$ when $\|\boldsymbol{\mu}\| < 1$, and $\theta < 0$ for $\|\boldsymbol{\mu}\| > 1$. Next we solve for $\boldsymbol{w}$, enforcing unit sum (but delaying non-negativity). With Lagrange multiplier $c$, we need to have

$$c = \frac{1}{2} \frac{\mu_k^2}{\left(\frac{w_k}{\theta} - 1\right)^2} \qquad \text{resulting in} \qquad w_k = \theta\left(1 + \sqrt{\frac{\mu_k^2}{2c}}\right).$$

Solving for the normalisation results in

$$c = \frac{1}{2}\left(\frac{\theta}{1 - \theta K}\|\boldsymbol{\mu}\|_1\right)^2 \qquad \text{whence} \qquad w_k = \theta + (1 - \theta K)\frac{|\mu_k|}{\|\boldsymbol{\mu}\|_1}.$$

Plugging this in, it remains to solve

$$\max_\theta \quad -\frac{\theta^2}{2(1 - \theta K)}\|\boldsymbol{\mu}\|_1^2 + \frac{\theta}{2}\left(1 - \|\boldsymbol{\mu}\|^2\right).$$

This concave problem is bounded by non-negativity of the right-hand side of (5). Cancelling the derivative results in a quadratic equation, with the single feasible solution

$$\theta = \frac{1}{K}\left(1 - \frac{\|\boldsymbol{\mu}\|_1}{\sqrt{K\left(1 - \|\boldsymbol{\mu}\|^2\right) + \|\boldsymbol{\mu}\|_1^2}}\right).$$

Filling this in yields the value and weights of the Lemma. Finally, we need to check for negativity in the weights. Using the above expression for the weights, have $\min_k w_k \geq 0$ if

$$-\frac{1}{\frac{\|\boldsymbol{\mu}\|_1}{\min_k |\mu_k|} - K} \leq \theta \qquad \text{i.e.} \qquad \frac{\|\boldsymbol{\mu}\|_1}{\sqrt{\|\boldsymbol{\mu}\|_1^2 - K\left(\|\boldsymbol{\mu}\|^2 - 1\right)}} \leq \frac{1}{1 - K\frac{\min_k |\mu_k|}{\|\boldsymbol{\mu}\|_1}}$$

which we can further reorganise to (5), as required. □

## H.4 Composition of two independent problems

We consider the case where we seek to answer two independent queries on disjoint sets of arms. Let $A, B$ be a partition of $[K]$. Suppose that the structure of the problem and the answers decompose according to this partition, i.e. $\mathcal{M} = \mathcal{M}^A \times \mathcal{M}^B$, $\mathcal{I} = \mathcal{I}^A \times \mathcal{I}^B$ and $i^*(\boldsymbol{\mu}) = i_A^*(\boldsymbol{\mu}^A) \times i_B^*(\boldsymbol{\mu}^B)$. Then we can also write all alternative sets $\neg i$ as $\neg i_A \times \mathcal{I}^B \cup \mathcal{I}^A \times \neg i_B$. It then holds that for all $\boldsymbol{w} \in \boldsymbol{w}^*(\boldsymbol{\mu}, \neg i)$,

$$\sum_{k \in A} w_k = \frac{\frac{1}{D(\boldsymbol{\mu}^A, \neg i_A)}}{\frac{1}{D(\boldsymbol{\mu}^A, \neg i_A)} + \frac{1}{D(\boldsymbol{\mu}^B, \neg i_B)}},$$

$$\sum_{k \in B} w_k = \frac{\frac{1}{D(\boldsymbol{\mu}^B, \neg i_B)}}{\frac{1}{D(\boldsymbol{\mu}^A, \neg i_A)} + \frac{1}{D(\boldsymbol{\mu}^B, \neg i_B)}},$$

$$\frac{1}{D(\boldsymbol{\mu}, \neg i)} = \frac{1}{D(\boldsymbol{\mu}^A, \neg i_A)} + \frac{1}{D(\boldsymbol{\mu}^B, \neg i_B)}.$$

Since the sample complexity is proportional to $1/D$ we obtain the natural conclusion that the number of samples needed to solve two independent queries is the sum of the samples needed by each query.