[Reviews · NeurIPS 2019]

Reviewer 1



The authors provides a novel and interesting look into the problem of pure exploration in the multiple-answers setting. The paper was clear in its exposition but the results could have been more insightful by comparing edge cases when this problem collapses to other well know problem. For example what happens in the best arm setting or the top-k setting? Some of the constants hide the ability to check and compare the results with the state of the art in these cases. In this paper you can derive bounds for an epsilon-good arm. Garivier and Kauffman (2016) have a lower-bound for such a scenario. Can you recover their lower bounds from your results? That is not clear.

Reviewer 2



This paper is technically sound. The theoretical analysis supports the claims in the paper. However, the main body of the paper is dense in terms of the notations and theoretical results, so it is not easy for me to follow. This paper contains a lot of great theoretical results, while I recommend the authors reorganize the paper to make it more readable. I have not seen the game-theoretic equilibrium argument presented in this paper, and I believe other researchers are likely to use this idea to prove the complexity results for more complicated problems.

Reviewer 3



Originality: This work proves tight bounds for the multi-answer pure exploration setting in the high-confidence regime, which remains open to the best of the reviewer's knowledge. The authors' approach is inspired by previous work on the single-answer setting, yet the work also contains sufficiently new techniques tailored to the challenges imposed by the multi-answer setting. Quality: I checked the proof sketches in the first eight pages as well as the proof of Theorem 1; they all look technically sound to me. Clarity: The paper is well-written and relatively easy to understand (considering the amount of technical details). Significance: This work derives asymptotically optimal bounds for the multi-answer setting, where prior approaches are shown to be not directly applicable due to the non-convexity introduced by the multi-answer assumption. This is a solid and important contribution to the field and also poses a few open questions for future work. *** added after author feedback *** I have read the rebuttal and my positive evaluation of this work remains unchanged.

[Author Response · NeurIPS 2019]

Dear Reviewers,

Thank you for your time and effort. We are happy to see the positive feedback, especially on the appeal of our
game-theoretic equilibrium technique.

**Reviewer 1** The main message concerning our algorithm in the single-answer case is that it is very close to Track-
and-Stop, and has similar performance.

The reviewer asks about the lower bound in (Garivier and Kaufmann, 2016). That lower bound deals with single-answer
problems exclusively, and the paper does not mention $\epsilon$-best arm. We believe the relevant paper is the earlier (Kaufmann,
Cappé and Garivier, 2015), which does have a lower bound for $\epsilon$-best arm in Remark 5.

As we show in our paper, the optimal rate at which the sample complexity grows with the confidence $\ln\frac{1}{\delta}$ is

$$\frac{1}{\min_{i\in i^*(\mu)}\max_{w\in\triangle}\inf_{\lambda\in\neg i}\sum_k w_k d(\mu_k,\lambda_k)}.$$

We can indeed obtain the bound from ours as follows. (Kaufmann, Cappé and Garivier, 2015) aim to get the best lower
bound obtainable from using alternative bandit models $\lambda\in\neg i$ which differ in only one arm, i.e. $\#\{k\mid\mu_k\neq\lambda_k\}=1$
(see Figure 1(a)). We search over all of $\neg i$. For $\epsilon$-Best Arm Identification specifically, it turns out that it suffices to
move only two arms (see Figure 1(b)).

Unfortunately, neither approach when taken to its extreme results in a closed form expression for the rate. For Gaussian
arms, it is known that the advantage of two-arm movements is at most a factor 2, see (Garivier and Kaufmann, 2016,
page 6). However, for Bernoulli bandits the advantage can be arbitrary.

**Reviewer 2** We can indeed cite `https://arxiv.org/abs/1905.08165` (note however that it was on arxiv for less
than a week before the submission deadline). It studies the general fixed confidence pure exploration problem with a
single answer, while the main point of interest of our paper is the multiple-answer case.

Thank you for including the typos you spotted. In contrast with the other reviewers, you found the paper dense. To
help us improve the paper, would you perhaps be able to update your review with line numbers of particularly opaque
paragraphs?

**Reviewer 3** Thanks you for your reference [1]. We will add it to the discussion as it is indeed relevant to our paper
for two reasons. It extends the Track-and-Stop approach to the general identification case (beyond Best Arm). Even
though the technical contribution in the paper are formulated for the Gaussian case only, the ideas naturally extend to
other families. In addition, the paper gives non-asymptotic terms that are hinting at problem complexity dependencies
beyond confidence. We are not currently making any claims on this front.

The paper matches the asymptotic rate up to a multiplicative constant. In our opinion though, this is a nice step on the
way to, but still a far cry from, matching the lower bound rate *exactly*. The latter is possible for Best Arm and, as we
show now, many other "general sampling" problems.

We have some ideas for and are are working on techniques for obtaining efficient algorithms non-asmptotic bounds
that matching the asymptotic optimal rate. It will be very interesting to compare lower-order terms once we are there.
However, this is a separate project beyond the current paper.

(a) Two single-arm motions that make arm $i=2$ incorrect. The blue moves arm $i$ to $\lambda_2=\mu_1-\epsilon$. The red moves any other arm $j\neq i$ to $\lambda_j=\mu_i+\epsilon$. All other arms $k$ remain fixed at $\lambda_k=\mu_k$ in either case.

(b) A single two-arm motion that makes arm $i=2$ incorrect. We move both arm $i$ to some level $\lambda_i=\zeta-\epsilon/2$ and some other arm $j\neq i$ to $\lambda_j=\zeta+\epsilon/2$. We choose the level $\zeta$ to optimise the bound.

[Meta-Review · NeurIPS 2019]

The reviewers all liked this paper, and so did I.